

# Spatially Resolved Infrared Radiofluorescence: Single-grain K-feldspar Dating using CCD Imaging

Dirk Mittelstraß[1] and Sebastian Kreutzer[2,3]

[1]Independent Researcher, Berthelsdorfer Str. 13, 09599 Freiberg, Germany
[2]Geography & Earth Sciences, Aberystwyth University, Wales, United Kingdom
[3]IRAMAT-CRP2A, UMR 5060, CNRS-Université Bordeaux Montaigne, Pessac, France

**Correspondence:** Dirk Mittelstraß (dirk.mittelstrass@luminescence.de) for technical questions and Sebastian Kreutzer (sebastian.kreutzer@aber.ac.uk) for questions on the experiments.

**Abstract.** The success of luminescence dating as a chronological tool in Quaternary science builds upon innovative methodological approaches, providing new insights into past landscapes. Infrared radiofluorescence (IR-RF) on K-feldspar is such an innovative method already introduced two decades ago. IR-RF promises considerable extended temporal range and a simple measurement protocol, with more dating applications published recently. To date, all applications use multi-grain measure-

ments. Herein, we take the next step by enabling IR-RF measurements on a single grain level. Our contribution introduces spatially resolved infrared radiofluorescence (SR IR-RF) on K-feldspars and intends to make SR IR-RF broadly accessible as a geochronological tool. In the first part of the manuscript, we detail equipment, CCD camera settings and software needed to perform and analyse SR IR-RF measurements. We use a newly developed *ImageJ* macro to process the image data, identify IR-RF emitting grains and obtain single-grain IR-RF signal curves. For subsequent analysis, we apply the statistical programming

environment *R* and the package 'Luminescence'. In the second part of the manuscript, we test SR IR-RF on two K-feldspar samples. One sample was irradiated artificially; the other sample received a natural dose. The artificially irradiated sample renders results, indistinguishable from conventional IR-RF measurements with the photomultiplier tube. The natural sample seems to overestimate the expected dose by ca 50% on average. However, it also shows a lower dose component resulting in ages consistent with the same sample's quartz fraction. Our experiments also revealed an unstable signal background due

to our cameras' degenerated cooling system. Besides this technical issue specific to the system we used, SR IR-RF is ready for application. Our contribution provides guidance and software tools for methodological and applied luminescence(-dating) studies on single grain feldspars using radiofluorescence.

## 1 Introduction

During the last two decades of advances in luminescence-based chronologies, two promising developments stand out but

somehow never took off: (1) Spatially resolved (SR) detection of optical- and thermoluminescence signals and (2) infrared radiofluorescence (IR-RF) of potassium feldspar (K-feldspar). Our perception is that the most significant obstacles in both approaches lie in imperfections of the available instrumentation and the complexity of the data analysis.





Although aware of this, we draw upon both developments, SR and IR-RF, and present a new approach: Spatially resolved infrared-radiofluorescence (henceforth: SR IR-RF) for measuring K-feldspar on a single grain level. This manuscript has two parts. After a brief literature review, the first part will outline the technical aspects and the data analysis methods. The second part will test and apply the developed approach.

As it concerns our manuscript's technical part, we attempt to summarise our work on SR IR-RF of K-feldspar carried out since 2015. We will present a detailed workflow and a new software toolchain and guidelines and technical suggestions like the parametrization of the used EM-CCD camera.

In the application part of our manuscript, we will present a first test of the hypothesis of whether SR IR-RF allows deciphering single feldspar grains' bleaching history. We used a sample from the Médoc area (south-west France) previously dated using non-spatially resolved IR-RF for this test.

## 1.1 Spatially resolved luminescence dating

Conventional luminescence readers rely on photomultiplier tubes (PMT) to detect luminescence emissions (e.g., Bortolot, 2000; Bøtter-Jensen et al., 2000; Richter et al., 2015; Maraba and Bulur, 2017). However, equivalent dose ($D_e$) distributions deduced from multiple-grain aliquots tend to scatter more than individual analytical uncertainties can explain (for a brief overview on various reasons, see Fitzsimmons, 2019) and a PMT does not allow to distinguish simultaneously emitted signals from individual grains.

Single-grain systems, such as employed by Bøtter-Jensen et al. (2003), use a laser to optically stimulate luminescence (OSL, Huntley et al., 1985) single grains sequentially. Hence, luminescence is collected grain wise. However, such a system does not suite when (1) the stimulation (heating, irradiation) can only be applied simultaneously to all grains, or (2) spatial mapping of the sample is desired (e.g., Duller and Roberts (2018)). For these reasons, luminescence imaging systems have been subject of research since, at least, the 1980s (cf. Huntley and Kirkley, 1985). In the 1990s, Coupled Charge Device (CCD) cameras became affordable and gained attraction for luminescence detection due to their high quantum efficiency in conjunction with a relatively simple technical implementation into existing systems. A variety of experimental and commercial image systems based on CCD cameras were developed (Duller et al., 1997; Spooner, 2000; Greilich et al., 2002; Baril, 2004; Clark-Balzan and Schwenninger, 2012; Chauhan et al., 2014; Greilich et al., 2015; Kook et al., 2015; Duller et al., 2020). However, the number of publications making use of those systems in the context of actual dating appears to be surprisingly small (e.g., Greilich et al., 2005; Rhodius et al., 2015; Duller et al., 2015).

Reasons for this lag of attention might be found in the technical complexity of luminescence imaging systems, combined with significant methodological issues such as image noise or signal cross-talk (Gribenski et al., 2015; Cunningham and Clark-Balzan, 2017). Thus, luminescence imaging techniques appear challenging to apply, and the efforts necessary to analyse the measurements might be considered disproportional to the scientific gain. To prevent spatially resolved IR-RF from failing for the same reasons, we intend to provide our software and methods as accessible, transparent and automated as possible.





## 1.2 The brief history of spatially resolved IR-RF dating

IR-RF dating applies ionising radiation to stimulate a fluorescence signal in K-feldspar at a wavelength of around $865\,\mathrm{nm}$ (Trautmann et al., 1999). This IR-RF signal decays with the accumulation of dose and resets through optical bleaching (e.g., a few hours to days of sunlight exposure). Determining ages up to ca $600\,\mathrm{ka}$ is reported in the literature (Wagner et al., 2010).

The early development of conventional non-spatially resolved IR-RF as dating technique was an effort of the group around Matthias Krbetschek at the TU Freiberg (Germany) (Trautmann et al., 1998, 1999; Krbetschek et al., 2000a; Erfurt and Krbetschek, 2003b). Their work cumulated in the infrared radiofluorescence single aliquot regenerative-dose (IRSAR) protocol (Erfurt and Krbetschek, 2003a). Although the IRSAR protocol is straightforward and promises extended temporal range, its invention had limited impact on the dating practice in Quaternary science (see Murari et al., submitted, for a detailed review).

One particular issue is the low bleachability of the IR-RF signal (at least two hours of natural sunlight, Trautmann et al., 1999), which potentially provokes partial bleaching effects. Other issues are potential inhomogeneities in the mineral composition and micro-dosimetry of the sample (Trautmann et al., 2000).

In the search for technical a solution, Krbetschek and Degering (2005)[1] conducted first spatially resolved (SR) IR-RF measurements on feldspars. In their experiment, the sample was irradiated from below with a $^{90}\mathrm{Sr}/^{90}\mathrm{Y}$ $\beta$-source. The RF signal was collected by a $45°$ mirror and a custom-made imaging optic, with the signal detected through an Electron Multiplying (EM)-CCD camera. Images were analysed using the software *AgesGalore* (Greilich et al., 2006).

When M. Krbetschek joined the Freiberg Instruments GmbH, the capability to perform SR IR-RF measurements became part of the design of the commercial available *lexsyg research* reader (Richter et al., 2013). Contrary to the original design by Krbetschek and Degering (2005), in this system, the $^{90}\mathrm{Sr}/^{90}\mathrm{Y}$ source is placed above the sample position. A circular opening in the middle of the source module enables luminescence detection (Richter et al., 2012). A sketch of this *lexsyg research* RF imaging module is shown in Fig. 1. With the early death of M. Krbetschek in 2012, the progression in the SR IR-RF technique's development came to a temporary hold.

## 2 Part I: Enabling spatially resolved radiofluorescence

In the following section, we outline technical aspects of relevance for successful SR IR-RF measurements. Although we were bounded to tailor some settings to a particular system, the overall parametrisation and the developed workflow is fairly system independent. More detailed information is available in the Appendix and the referenced resources.

### 2.1 Equipment

All measurements presented in this manuscript were performed on a single Freiberg Instruments *lexsyg research* reader (Richter et al., 2013) at the IRAMAT-CRP2A in Bordeaux (reader name 'L2'). The system is equipped with a ring-shape type $^{90}\mathrm{Sr}/^{90}\mathrm{Y}$

---

[1]These results were never formally published. However, we are happy to share their presentation on the 11[th] International LED conference (2005, Cologne, Germany) upon request.





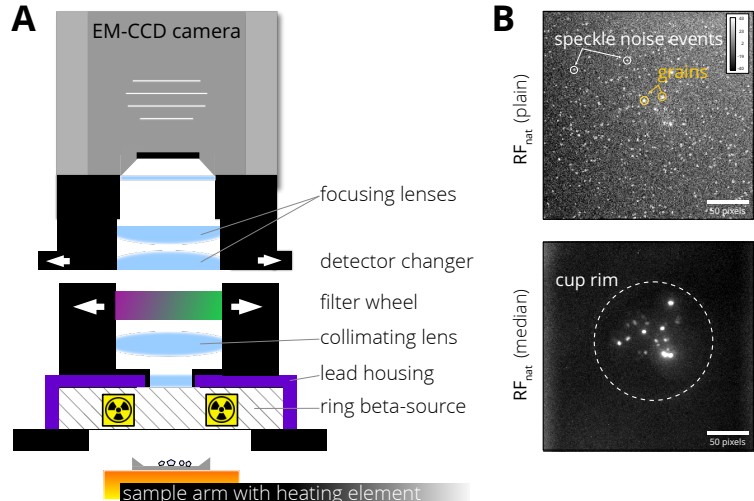

**Figure 1.** (A) Technical sketch of the camera system for spatially resolved IR-RF measurements at the *lexsyg research* reader 'L2' in Bordeaux. The collimating lens position and the camera's height were adjusted manually to obtain the best IR-RF image quality. (B) Typical image output of a natural IR-RF image stack. The upper picture shows an unprocessed but background-corrected SR IR-RF image taken with High SNR setting (see Table 2). Two speckle noise events caused by bremsstrahlung (white) and two grain IR-RF signals (yellow) are marked. Individual grain signals are hardly distinguishable from the image noise. The lower image shows the processed image stack's median image, where light from individual grains is visible. The dashed white line marks the rim of the sample carrier (here stainless-steel cup).

$\beta$-source (Richter et al., 2012) delivering ca $3.5\,\mathrm{Gy\,min^{-1}}$ to K-feldspar grains with a size of $125 - 250\,\mu\mathrm{m}$ (cf. Frouin et al.,
85 2017).

For luminescence detection, we used a *Princeton Instruments ProEM: 512B+ eXcelon* camera with a $512 \times 512$ pixel unichromatic CCD sensor sitting on an automated detector changer. The camera has a quantum efficiency (QE) of $\geqslant 80\%$ between $450\,\mathrm{nm}$ and $750\,\mathrm{nm}$. At the K-feldspar IR-RF emission centred at ca $865\,\mathrm{nm}$ the QE is around $60\%$. For the IR-RF measurements, we placed a *Chroma D850/40x* interference filter between $\beta$-source and CCD camera. The custom-made optic
90 has a numerical aperture (NA) of about 0.2 and a lateral magnification of 0.6, leading to an image resolution of $27\,\mu\mathrm{m}$ per pixel. Prior to our experiments, we adjusted the optical focus by manually calibrating the camera's installation height until we obtained the best image quality in terms of sharpness and minimised distortion.

The system was equipped with a solar light simulator (SLS) system facilitating LEDs with broad peaks centred at $365\,\mathrm{nm}$, $462\,\mathrm{nm}$, $523\,\mathrm{nm}$, $590\,\mathrm{nm}$, $625\,\mathrm{nm}$ and $850\,\mathrm{nm}$ (Richter et al., 2013). The system is the same as used for the experiments by
95 Frouin et al. (2015) and Frouin et al. (2017).

However, over the years, the system received a couple of hardware upgrades tackling various problems (cf. Kreutzer et al., 2017). In 2018, an improved drive train for the sample arm as well as modernised control hardware in conjunction with a new



100 W at 48 V $Si_3N_4$ heater controlled by a PT1000 thermocouple was installed (pers. communication Freiberg Instruments GmbH).

## 2.2 Software

Our software toolchain consisted of three different tools: *LexStudio2* for measurement sequence control, *ImageJ* for image processing and the R function library *Luminescence* for data analysis. *ImageJ* and the *Luminescence* package are open-source (GPL-3 licence) and freely available for all major platforms (*Windows*, *Linux*, *macOS*). However, our software toolchain was tested so far just on *Windows 10* and *macOS* (v10–v11). Detailed installation guides and additional download links to the SR IR-RF specific software modules can be found at https://luminescence.de.

### 2.2.1 Image acquisition with *LexStudio2*

We used the software *LexStudio2* (version 2.5.0, 2019-11-01) shipped with the measurement system for sequence writing, camera parametrisation, and image acquisition. For the presented work, Freiberg Instruments updated *LexStudio2* in 2018/2019 with a new module to control the camera settings relevant for luminescence imaging (Fig. 2A). The new module also enables sequence-synchronous camera control and data handling. Thus, sequence writing differs not from routine luminescence measurements with a PMT. The new *LexStudio2* camera module uses the 32-bit *PVCAM* drivers by Princeton Instruments and maintains the compatibility to the camera control software *WinView* shipped with the ProEM camera. Unfortunately, this enhanced version of *LexStudio2* is currently bound to 32-bit Microsoft Windows platforms. The obtained data, however, can be separately processed and analysed on other computers with other platforms. The data consists of one image stack for each RF measurement, saved as a 16-bit greyscale TIF file. To prevent system crashes due to the *3 GB barrier* of 32-bit platforms, *LexStudio2* provides an option to split large data sets automatically.

### 2.2.2 Image processing with *ImageJ*

For processing the image data, we used the open-source software *ImageJ* (version: 1.52p) (Schneider et al., 2012). We developed a macro called "SR-RF" (file `SR-RF.ijm`, see supplement) to automatise the workflow. The SR-RF macro is a plain ASCII-file and written in the JavaScript-like *ImageJ* macro language. It provides a graphical user interface (Fig. 2B) to simplify user-interactions. The output is an ASCII text file with the file-ending `*.rf`. The file contains the single-grain IR-RF curves, the size and spatial location of the associated regions of interests (ROIs) and further image processing information. We used the enhanced *ImageJ* distribution *Fiji* (https://fiji.sc) (version: 2.0.0-rc-69) for most of our analyses. A cross-platform version of *ImageJ* and the SR-RF macro and all necessary plug-ins pre-installed can be downloaded from https://luminescence.de. A short description of how to install the SR-RF macro and its dependencies and detailed documentation of the macro can also be found on our website. Interfacing of the macro to other programs is possible through the additionally supported *ImageJ* *batch-mode*.



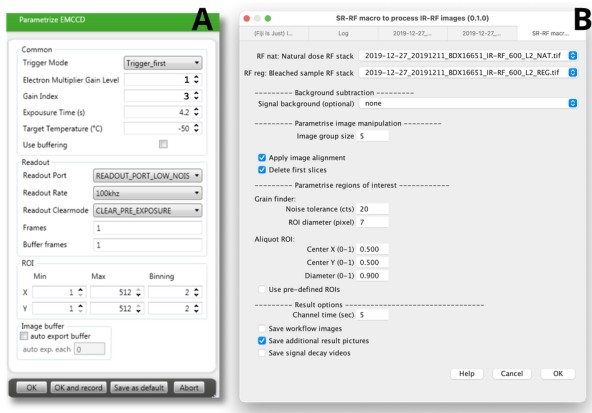

**Figure 2.** Screenshots of (A) the *LexStudio 2* interface to parametrise the CCD camera, and (B) the ʻSR-RFʼ *ImageJ* marco interface to analyse IR-RF images.

### 2.2.3 Data analysis with R

We employed the statistical programming environment R (R Core Team, 2020) and the package 'Luminescence' (Kreutzer
et al., 2012, 2020) for processing the IR-RF single-grain data.

Therefore, we developed two new functions for a seamless data import and processing of `*.rf` files:
`Luminescence::read_RF2R()` and `Luminescence::plot_ROI()` ('Luminescence' $\geq$ v0.9.8). Both functions work
in conjunction with the already available function `Luminescence::analyse_IRSAR.RF()`. See below for an applica-
tion example.

Advanced users can also deploy our experimental R package dedicated to spatially resolved luminescence data analysis
called 'RLumSTARR' (Kreutzer and Mittelstrass, 2020a). The sole relevance of 'RLumSTARR' for this contribution is the
function `RLumSTAAR::run_ImageJ()`. We used this function to run *ImageJ* in a batch mode and auto process our image
data. However, 'RLumSTARR' is not required to analyse SR IR-RF data.

### 2.3 Measurement protocol

We applied the $RF_{70}$ single aliquot protocol by Frouin et al. (2017), an improved version of the IRSAR protocol (Erfurt and
Krbetschek, 2003a). The $RF_{70}$ sequence (Table 1) includes two IR-RF measurements: One for the natural signal ($RF_{nat}$) and
one for the regenerated signal ($RF_{reg}$). In the data analysis process, the $RF_{nat}$ signal curve is slid vertically and horizontally
signal curve until to best match the $RF_{reg}$ curve. The horizontal sliding distance is the accumulated dose needed to match the
natural RF signal, thus defining the equivalent dose (Murari et al., 2018). Measurement durations are user-defined. However,
$RF_{reg}$ should be longer than the sample's expected natural dose. $RF_{nat}$ should not contain less than 70 data points (in our case
images) to give sufficient statistical confidence when using the sliding method (Frouin et al., 2017, their supplement, proposed
at least 40 channels for a resolution of 15 s/channel).





We used the same solar simulator settings for the bleaching as Frouin et al. (2015) but with increased UV stimulation power: 365 nm: $20\,\mathrm{mW\,cm^{-2}}$, 462 nm: $61\,\mathrm{mW\,cm^{-2}}$, 525 nm: $53\,\mathrm{mW\,cm^{-2}}$, 590 nm: $37\,\mathrm{mW\,cm^{-2}}$, 623 nm: $112\,\mathrm{mW\,cm^{-2}}$, 850 nm:

$94\,\mathrm{mW\,cm^{-2}}$.

**Table 1.** Applied IR-RF measurement sequence according to the $RF_{70}$ protocol by Frouin et al. (2017).

| # | Step | Treatment | Measurement |
|---|------|-----------|-------------|
| 1 | Shallow trap depletion | Preheat at 70 °C for 900 s | - |
| 2 | Natural dose IR-RF | $\beta$-irradiation at 70 °C for `<user defined time>` | $RF_{nat}$ |
| 3 | Signal resetting | Bleaching with in-built solar simulator for 3 h | - |
| 4 | Wait for phosphorescence to decay | Pause for 1 h | - |
| 5 | Shallow trap depletion | Preheat at 70 °C for 900 s | - |
| 6 | Regenerative dose IR-RF | $\beta$-irradiation at 70 °C for `<user defined time>` | $RF_{reg}$ |

## 2.4   Camera settings

While the enhanced *LexStudio2* version automates image acquisition, it does not free the user from parametrising the camera. In the following, we will advise on the most relevant camera settings and their impact on image noise and signal sensitivity. We derive parts of our advice from signal-to-noise ratio (SNR) estimations summarised in Appendix A. Table 2 lists major

correlations between CCD camera settings and data quality. Table 3 lists the camera settings we used in our experiments. For more in-depth insights into the scientific CCD camera technology we may refer to Janesick (2001) and the *Andor Learning Centre* (https://andor.oxinst.com/learning/; search for "Andor Academy").

### 2.4.1   Set the CDD chip temperature low, but not too low

One primary source of image noise arises from the dark current of the CCD chip. The dark current is highly temperature-

dependent. Cooling the CCD chip temperature by $10\,\mathrm{K}$ quarters the dark current related signal background and halves the dark current related image noise (Fig. S1). Thus, it seems obvious to set the target CCD temperature as low as the cooling system allows. However, we strongly recommend setting the target temperature between $10\,\mathrm{K}$ and $15\,\mathrm{K}$ above the theoretical minimum. An RF measurement takes hours, enough time for the camera electronics to warm up, or change the system temperature. The resulting fluctuations in the CCD temperature induce changes in the background signal level during the RF measurement.

These background instabilities are hard to correct in the post-processing. Therefore, a stationary CCD temperature level is mandatory and eased by leaving the cooling system enough headroom for corrections.





### 2.4.2 Select a slow readout rate, but not too slow

The CCD chip readout process induces another source of image noise called read noise or readout noise (cf. standard textbooks for both notations). Longer exposure times lead to better SNR because more signal is gathered while the readout noise remains
constant.

Another way to reduce readout noise is to choose a slow CCD readout rate. In our system, the slowest available readout rate is $100\,\mathrm{kHz}$. At this rate, a full resolution ($512 \times 512\,\mathrm{px}$) CCD readout takes $2.13\,\mathrm{s}$. If the readout process lasts longer than the RF measurement channel, either images are lost, or the camera runs asynchronous to the measurement sequence. In our systems, the readout process started after the preset time interval for the image exposure ended. The camera is then locked until
the image data is transferred to the computer. Thus, the user has to incorporate a camera dead time when parametrising channel width and exposure time in the camera's sequence settings, see Table 3 and Appendix A. However, all modern scientific CCD cameras, including our *ProEM* camera, can read out the last image while already gathering signal light for the next image. The camera dead time is an setting particular to the *LexStudio2* software solution we used. Later software iterations or more advanced systems might set exposure time and channel width equal by default.

### 2.4.3 Do not use EM gain

EM-CCD cameras have an electron-multiplying (EM) register that amplifies the detected signals above readout noise if activated. The EM-mode allows for highly sensitive high-frame-rate imaging, but it comes at costs. (1) It induces image noise (exceed noise), (2) it reduces the dynamic range and the linearity of the signal acquisition, (3) it amplifies dark current signals and thus dark noise, and (4) it amplifies local pixel over-exposures leading to pixel-well overflows. Especially the last point
is problematic for RF imaging. The nearby $^{90}\mathrm{Sr}/^{90}\mathrm{Y}$ $\beta$-source shielding emits secondary X-ray photons (bremsstrahlung, cf. Liritzis and Galloway, 1990), which induce localized high signal events at the CCD chip upon impact. The result is images speckled with bright spots (see Fig. 1B). If this speckle-noise gets amplified by the EM-mode, it causes streaks on the image which are hard to remove by image processing algorithms.

### 2.4.4 Consider hardware pixel binning

The most straightforward approach to improving the SNR and the signal sensitivity is pixel binning performed by the CCD camera image processing software like *ImageJ*. This software pixel binning, however, is less effective than potential hardware binning by the camera. With applied hardware pixel binning, multiple pixels are considered as one pixel and read out together. This feature reduces the readout noise per imaged area and reduces the readout time and therefore the camera dead time (if applicable). As a side effect, the image stacks' file size is also reduced, positively impacting image processing time. We applied
$2 \times 2$ pixel binning as a default setting and deactivated it only if we had sufficiently bright samples. On the downside, pixel binning lowers the camera resolution to $256 \times 256$ pixel, corresponding to a decreased spatial resolution of $54\,\mu\mathrm{m}$ (before $27\,\mu\mathrm{m}$).





**Table 2.** Correlation between basic camera settings and data quality. Up arrows: Increasing this parameter leads to an increase of, e.g., noise, time span, intensity. Down arrow: Increasing this parameter leads to a decrease of the corresponding attribute. Right arrow: Changed parameter settings do not affect the attribute.

| Camera setting | Source of information loss | | | Data quality | |
| --- | --- | --- | --- | --- | --- |
| | Readout noise | Dark noise | Dead time | Signal per | SNR |
| | per ROI and image | per ROI and Image | per image[*] | pixel | |
| Exposure time | $\Rightarrow$ | $\Uparrow$ | $\Rightarrow$ | $\Uparrow$ | $\Uparrow$ |
| Readout rate | $\Uparrow$ | $\Rightarrow$ | $\Downarrow$ | $\Rightarrow$ | $\Downarrow$ |
| Pixel binning | $\Downarrow$ | $\Rightarrow$ | $\Downarrow$ | $\Uparrow$ | $\Uparrow$ |
| CCD temperature | $\Rightarrow$ | $\Uparrow$ | $\Rightarrow$ | $\Rightarrow$ | $\Downarrow$ |

[*]camera dead time occurs only if a sequential CCD chip readout mode is applied (Full Frame mode)

**Table 3.** Recommended settings for a Princeton Instruments *ProEM512* camera employed in a Freiberg Instruments *lexsyg research* system.

| Camera setting | High SNR (default) | Full resolution |
| --- | --- | --- |
| Channel width | 5 s | 5 s |
| Exposure time | 4.15 s | 4.5 s |
| Readout rate | 100 kHz | 1 MHz |
| Pixel binning | 2 x 2 | off (1 x 1) |

## 2.5 Image processing

We obtain two $\ast$`.tif` files respectively image stacks (a series of images per file) per aliquot from the $RF_{70}$ protocol (Table 1).
Both image stacks are affected by speckle noise caused by bremsstrahlung. Besides, the $RF_{reg}$ images might be displaced or rotated compared to the $RF_{nat}$ images due to uncertainties in the aliquot positioning (Kreutzer et al., 2017).

Both issues and the grain identification, are addressed by the SR-RF macro in *ImageJ*. The image processing has four steps (Fig. 3): (1) Speckle noise is removed, (2) both image stacks are geometrically aligned, (3) individual grains are identified, and (4) single grain RF curves are extracted. Table 4 gives recommendations for the macro settings. For more details on the
deployed *ImageJ* commands, we refer to our SR-RF macro documentation on https://luminescence.de as well as the *ImageJ user guide* by Ferreira and Rasband (2012).

### 2.5.1 Step 1: Median filter

We used the *ImageJ* command `Grouped Z Project` (Ferreira and Rasband, 2012) to erase bremsstrahlung's spots. The images of both image stacks are grouped in quantities according to the user-defined parameter `Group Size`. Each group's





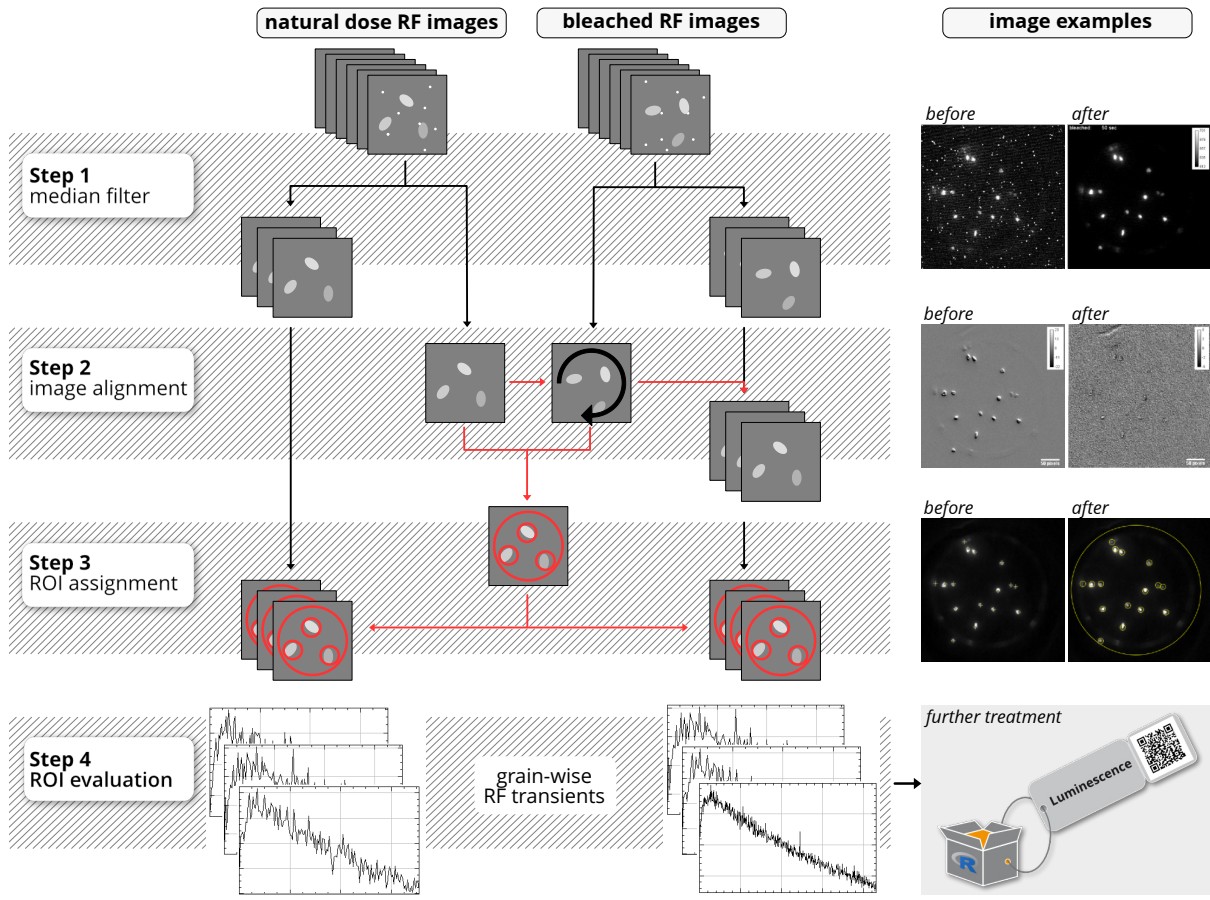

**Figure 3.** Image processing workflow as performed by the *ImageJ* macro *SR-RF.ijm*.

images are combined to one image by taking the median pixel value for each pixel location. This process removes signal outliers while maintaining the fundamental shape of the signal curve (Velleman, 1980). Speckles caused by bremsstrahlung occur in random locations. Hence, it is unlikely that the same pixel is affected more than once during a time interval related to the measurement of just a few images. The statistical likelihood of surviving speckles increases with longer image exposure times but decreases with larger group sizes. For the measurement system we used, and with an exposure time of 5 s, a group

size of five is sufficient to eradicate speckle noise.

### 2.5.2 Step 2: Image alignment

We used the *ImageJ* plug-in `TurboReg` by Thévenaz et al. (1998) to detect and correct aliquot movement. It is the same algorithm used by Greilich et al. (2015). The $RF_{nat}$ and the $RF_{reg}$ stack are aligned by comparing their global median images. Equal to other regression algorithms, the differences between median images are summed up to one residual value.




The $RF_{reg}$ median image is rotated and translated until the minimum is found. The rotation and translation parameters are then applied to all images of the $RF_{reg}$ stack.

To interpolate the signals for the fine movement of the alignment, *ImageJ* offers three methods `none`, `bilinaer` and `bicubic`. We tested the interpolation methods for sample TH0 (see below, Sec. 3.4.1) and selected `bicubic` as hidden preset value.

### 2.5.3 Step 3: Grain detection and ROI assignment

We used the *ImageJ* command `Find Maxima` (Ferreira and Rasband, 2012) to identify individual mineral grains. As reference image serves the arithmetic mean image of the two median images from step 2 (Sec. 2.5.2). There, the `Find maxima` algorithm searches for local maxima in the pixel values. The user-defined parameter `Noise tolerance` controls the algorithm's sensitivity, which defines how much higher than the surrounding area a pixel value must be. A higher `Noise tolerance` value leads to higher robustness against optical reflections and signal outliers but a lower grain detection likelihood. A circular ROI is assigned to each local maximum. The diameter of these circles is user-defined through the `ROI diameter` parameter.

### 2.5.4 Step 4: Extract single RF curves

We used the *ImageJ* `ROI manager` to obtain the arithmetic mean of the pixel values in each ROI for each image in the $RF_{nat}$ stack and $RF_{reg}$ stack. Thus, the consecutive average signal in one ROI forms the IR-RF curve of one sample grain. These single grain IR-RF measurements and the lateral position of each ROI, are saved into one ASCII text file (`table.rf`) to be further analysed with other software than *ImageJ*.

**Table 4.** Recommended SR-RF macro settings for the first try, depending on the sample brightness and the grain size. Parameter refinements depending on the system and the sample might be necessary.

| Macro parameter | $\sim 50 - 80\,\mu m$ **grain size** | | $\sim 180 - 250\,\mu m$ **grain size** | |
| --- | --- | --- | --- | --- |
| | Bright sample at full resolution[*] | Dim sample at high SNR setting[*] | Bright sample at full resolution[*] | Dim sample at high SNR setting[*] |
| Image group size | 5 | 5 | 5 | 5 |
| Noise tolerance | 15 | 10 | 15 | 10 |
| ROI diameter | 5 | 3 | 12 | 7 |

[*]refers to the recommended camera settings in Table 3.





## 2.6 Single grain data analysis in R

We analysed the single grain IR-RF data the same way that we would analyse conventional PMT IR-RF measurements. A
simple R script to analyse the `table.rf` file of one aliquot reads as follows (R package 'Luminescence'):

```
#load R package 'Luminescence'
library(Luminescence)

#import data
file <- file.choose()
RF_data <- read_RF2R(file)

#get ROI locations (optional)
ROI_data <- plot_ROI(RF_data)

#determine equivalent doses
equivalent_doses <- analyse_IRSAR.RF(
  object = RF_data,
  method = "SLIDE",
  method.control = list(
    vslide_range = "auto",
    correct_onset = FALSE))

#plot dose distribution
plot_AbanicoPlot(equivalent_doses)
```

Here, the new function `read_RF2R()` converts the `table.rf` file into a list of `RLum.Analysis` objects. Each `RLum.Analysis`
object contains the $RF_{nat}$ and $RF_{reg}$ curves of one ROI. The equivalent dose of each ROI is calculated by `analyse_IRSAR.RF()`
which was already introduced and used by Frouin et al. (2017). The resulting dose distribution can be displayed and further
evaluated by any of the various functions for dose statistics, the 'Luminescence' package provides. In the example above, we
allowed vertical sliding after Murari et al. (2018) in the function `analyse_IRSAR.RF()` (parameter `vslide_range`).
Vertical sliding can improve the equivalent dose results' accuracy but need a significant curvature in the IR-RF decay to work
properly. Vertical sliding can be deactivated by setting `vslide_range = NULL` or removing the parameter. The function
`plot_ROI()` displays and returns the ROI locations and returns the euclidean distance between them. This information is
useful to study the impact of signal cross-talk.

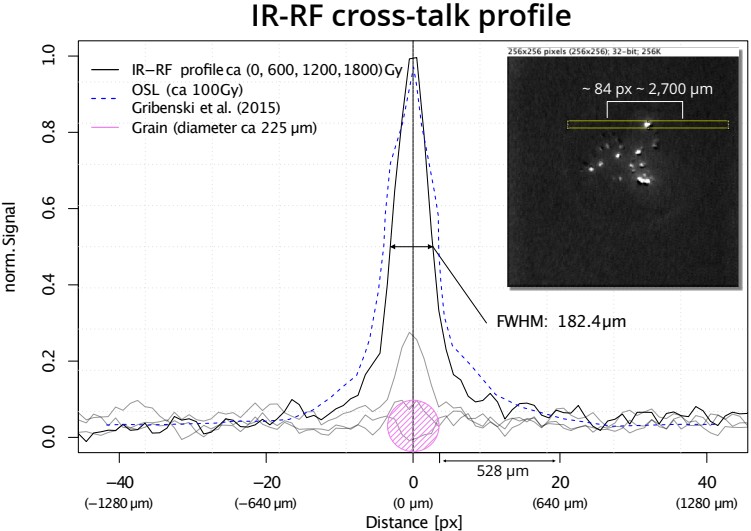

**Figure 4.** IR-RF cross-talk profile of one single grain. The inset shows the *ImageJ* image with the rectangle area selected for the profiling over ca 1,800 Gy (along the all image slices of the image stack). The solid black lines show the IR-RF signal. For illustrative reasons, we show only a few curves. The dashed red line shows the approximated OSL cross-talk profile recorded by Gribenski et al. (2015) after a dose of ca 100 Gy. The blue shaded area approximates the grain (not in height but width). Please note that the data by Gribenski et al. (2015) were only added to provide a rough qualitative comparison. Please note that the metric distances (sub-labels) refer to the chip surface.

## 2.7 Signal cross-talk

An issue in OSL and TL imaging flagged by Gribenski et al. (2015) and further discussed by Cunningham and Clark-Balzan (2017) is signal cross-talk. Two independent effects cause signal cross-talk: (1) signal light misdirected by optical aberrations and (2) signal light backscattered by the silicone fixation layers and the sample carrier's surface. The visual perceptions are blurry luminescence images and signal halos around individual grains. These halos can extend into the ROIs of near located other grains. This cross-talk effect blends the IR-RF curves and potentially narrows the $D_e$ distribution.

Gribenski et al. (2015) investigated the effects of signal-cross talk in spatially resolved OSL measurements. They found a substantial effect on the equivalent dose outcome and supposed optical aberrations as the primary signal cross-talk source. Like us, they used a *lexsyg research* device equipped with a *ProEM512B* camera. However, they measured at the OSL/TL sample position equipped with a custom-made multi-purpose optic (Richter et al., 2013; Greilich et al., 2015). The OSL/TL optic was designed with large opening angles and high UV-to-NIR transmittance. This design decision enabled a maximum of signal yield for various applications but counteracted the optical correction of spherical and chromatic aberration and their secondary effects like astigmatism.

The optic of the RF position has a far smaller aperture ($NA_{RF} \approx 0.2$ vs $NA_{OSL/TL} \approx 0.5$) and therefore less spherical aberration. In addition, we consider chromatic aberration as negligible because we performed the focus calibration and all





measurements at the same wavelength (865 nm). As Fig. 4 shows, the effect of signal cross-talk appears to be weaker in our measurements as observed by Gribenski et al. (2015) and should be insignificant for inter-grain distances above ca 500 µm. We tried to maintain this distance by preparing our samples with a very low grain density.

Nevertheless, for samples with a high grain density or a high grain intensity inhomogeneity, signal cross-talk will be an issue. We propose the following countermeasures to reduce the effect; none of them applied in our experiments though:

– **Special sample carriers**: punched or black or polished to minimize backscattered luminescence light

   – **Improved optics**: A more specialized optic (the lenses are exchangeable) can further reduce spherical aberration

   – **Application of mathematical corrections**, such as suggested by Cunningham and Clark-Balzan (2017)

## 3   Part II: Testing spatially resolved radiofluorescence

### 3.1   Samples

We selected two potassium bearing (K-feldspar) samples to apply and test our SR IR-RF tools and their settings. The first sample (TH0, grain size: $125 - 250$ µm) is a modern analogue sample of aeolian origin from the Sebkha Tah in Morocco (cf. Bouab, 2005). It is the same sample used by Frouin et al. (2017) to calibrate the $^{90}$Sr/$^{90}$Y source of the very same reader used for our measurements here. The sample was exposed to a $\gamma$-dose of $56.02$ Gy ($c_v$~2%) in 2015 (Frouin et al., 2017). The second K-feldspar sample (BDX16651, grain size: $100 - 200$ µm) originates from a coastal dune in the Médoc area (south-

west France). For this sample, Kreutzer et al. (2018) estimated a palaeodose of $50.7 \pm 5.7$ Gy for the quartz fraction and $96.2 \pm 8.0$ Gy for the here measured K-feldspar fraction. For details on the sample preparation procedures, we refer to the cited literature.

### 3.2   Experiments

TH0 allowed us to calibrate the $^{90}$Sr/$^{90}$Y source with SR IR-RF and compare the results with the PMT's calibration measure-

ments. For this experiment, the detector changed in alternating turns, i.e. after measuring an aliquot with the PMT, another aliquot was measured using the EM-CCD camera, and then we measured an aliquot again with the PMT, and so on. We tested whether both measurements estimate statistically indistinguishable source dose rates.

The measurements of BDX16651 aimed at one main application of single grain measurements: differentiation between grain fractions with different bleaching history. Kreutzer et al. (2018) reported an age of $37.0 \pm 4.9$ ka for the feldspar fraction and

$26.1 \pm 3.5$ ka for the quartz fraction. While both ages overlap within $2\sigma$, Kreutzer et al. (2018) reported consistently older ages for the feldspar fraction compared to the quartz fraction for all samples from the site. Therefore, they argued that the natural bleaching was likely insufficient to reset the IR-RF signal of the feldspar grains. SR IR-RF should confirm the quartz result obtained by Kreutzer et al. (2018) and potentially enable us to identify those grains that received a full signal resetting before the last burial.




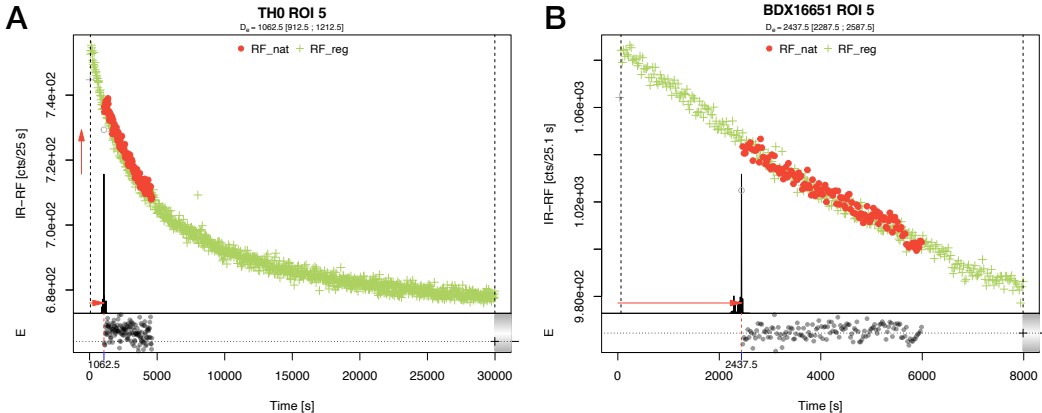

**Figure 5.** Typical IR-RF curves for the sample TH0 (left) and BDX16651 (right). Both curves were extracted from ROIs following the procedure outlined in the first part of the manuscript. For determining the $D_e$ the sliding method was used. Due to the absence of any curvature in the observed dose range, no vertical sliding was applied to sample BDX16651 measurements (B). *SR-RF* macro settings as follows: image group size: 5, noise tolerance: 20 (TH0), 30 (BDX16651), grain diameter: 7.

For both samples, feldspar grains were dispersed randomly on stainless-steel cups aiming at a low grain density. However, no mask or other aid was used because this reflects are more realistic aliquot preparation procedure in most laboratories. We aimed at 30 to 50 grains per aliquot. We prepared at least three cups per sample. Irradiation times were equal to values reported in Frouin et al. (2017) and Kreutzer et al. (2018): $3,600\,\mathrm{s}$ ($RF_{nat}$) and $30,000\,\mathrm{s}$ ($RF_{reg}$) for sample TH0; $3,600\,\mathrm{s}$ and $10,000\,\mathrm{s}$ for sample BDX16651.

Figure 5 shows typical IR-RF curves from one ROI (in our case one grain) for TH0 (Fig.5A) and BDX16651 (Fig.5B). To obtain the $D_e$s, we applied the vertical and horizontal sliding technique (Murari et al., 2018) to TH0 and used only horizontal sliding for sample BDX16651 due to the absence of visible curvature in the IR-RF curve.

As rejection criteria, we applied the default test criteria (cf. Frouin et al., 2017, their supplement) of the function `analyse_IRSAR.RF()`. Two of those criteria were of relevance for our contribution: `curves_ratio` and `curves_bounds`.

The first calculates the ratio of $RF_{nat}$ over $RF_{reg}$ in the range of $RF_{nat}$. If it exceeds a certain threshold (here 1.001) it usually indicates that the $RF_{nat}$ best matched the $RF_{nat}$ while lying above the $RF_{reg}$ (additionally confirmed by visual inspection), violating the assumption that the highest IR-RF signal is observed for the $RF_{reg}$ after bleaching. If the second, `curves_bounds`, criterion is flagged, the $RF_{nat}$ cannot match the $RF_{reg}$ within the measured range of $RF_{reg}$; an observation usually made for very noisy, flat curves.

The raw data of our measurements, along with the applied R scripts and partially pre-processed are available open-access (Kreutzer and Mittelstrass, 2020b).





### 3.3 Technical camera issues

While we measured at least three cups with grains per sample, the number of usable cups presentable here finally narrowed down to one cup each. A malfunction in the cooling system of our camera stopped us from conducting more measurements.

This cooling system degraded over the last five years, continually increasing the lowest reachable temperature from at least $-70\,°C$ in 2015 to about $-45\,°C$ in 2019. As we already mentioned above, the CCD chip temperature's spatial and temporal uniformity is necessary to ensure a stable and homogeneous signal background. We provide additional insights Appendix A2 and Figs. S1 and S2 in the supplement. Unfortunately, our camera's cooling system finally lost its ability to maintain stable chip temperatures during our measurements meant for publication. Significant variations in the IR-RF curves' signal background

required us to discard most of our measurements (see Fig. S4 for an example).

Independently from this issue, we further discarded grains located at the rim of the stainless-steel cup, where our analysis indicated exceeded IR-RF curve boundaries for unknown reasons (i.e. no match between $RF_{nat}$ and $RF_{reg}$).

### 3.4 Results

Figure 6 illustrates the final results for the two remaining cups. One for TH0 (Fig.6A, upper part) and one for BDX16651

(Fig.6B, lower part). For each sample, we show an image taken with the camera (left-hand side) during the measurements and an Abanico plot (Dietze et al., 2016) of the distribution of the results. ROI pixels (diameter 7 px, see Table 4) taken for the $D_e$ analysis are coloured green and numbered. The numbers are displayed again in the Abanico plots (right-hand side). The results of TH0 display dose rates in $Gy\,s^{-1}$ and equivalent doses in $Gy$ for BDX16651. We applied the average dose model (Guérin et al., 2017) to both distributions with an assumed $\sigma_m$ of 0.05.

### 330 3.4.1 TH0

SR RF-RF measurements of sample TH0 on 10 grains (ca 20 grains on the cup, 11 grains emitted sufficient light for the analysis, one grain discarded) obtained a source-dose rate of $0.055 \pm 0.004\,Gy\,s^{-1}$ (date: 2019-09-13). This value is consistent with the source-dose rate calibration value obtained through conventional IR-RF PMT measurements with the same sample (Fig. S5, measurement date: 2019-09-13, n = 10, $0.056 \pm 0.001\,Gy\,s^{-1}$). Hence, it confirms our hypothesis that the calibration results

obtained through SR IR-RF and IR-RF PMT measurements are indistinguishable. Furthermore, it gives some confidence that these measurements where not effect by the cooling-system malfunction for the camera.

We further tried to determine to which extent the results depend on chosen ROI size (here diameter 7 px) and the interpolation method used to correct the image for translation and rotation (Sec. 2.5.2). As interpolation method, we obtained best results for the option `bicubic` (see Fig. S3) which is the default in the SR-RF *ImageJ* macro `SR-RF`. The ROI diameter should mimic

the approximated grain size or be a bit larger (see also Fig. S3). We observed a plateau of results for ROI sizes between 5 px and 10 px for sample TH0. Smaller values should not be selected because the ROI finding algorithm may not reliably select the grain centre. For larger values, signal cross-talk effects likely become an issue. Although the median appears to be rather robust for all ROI sizes between 5 px and 30 px for `bicubic` (Fig. S3).



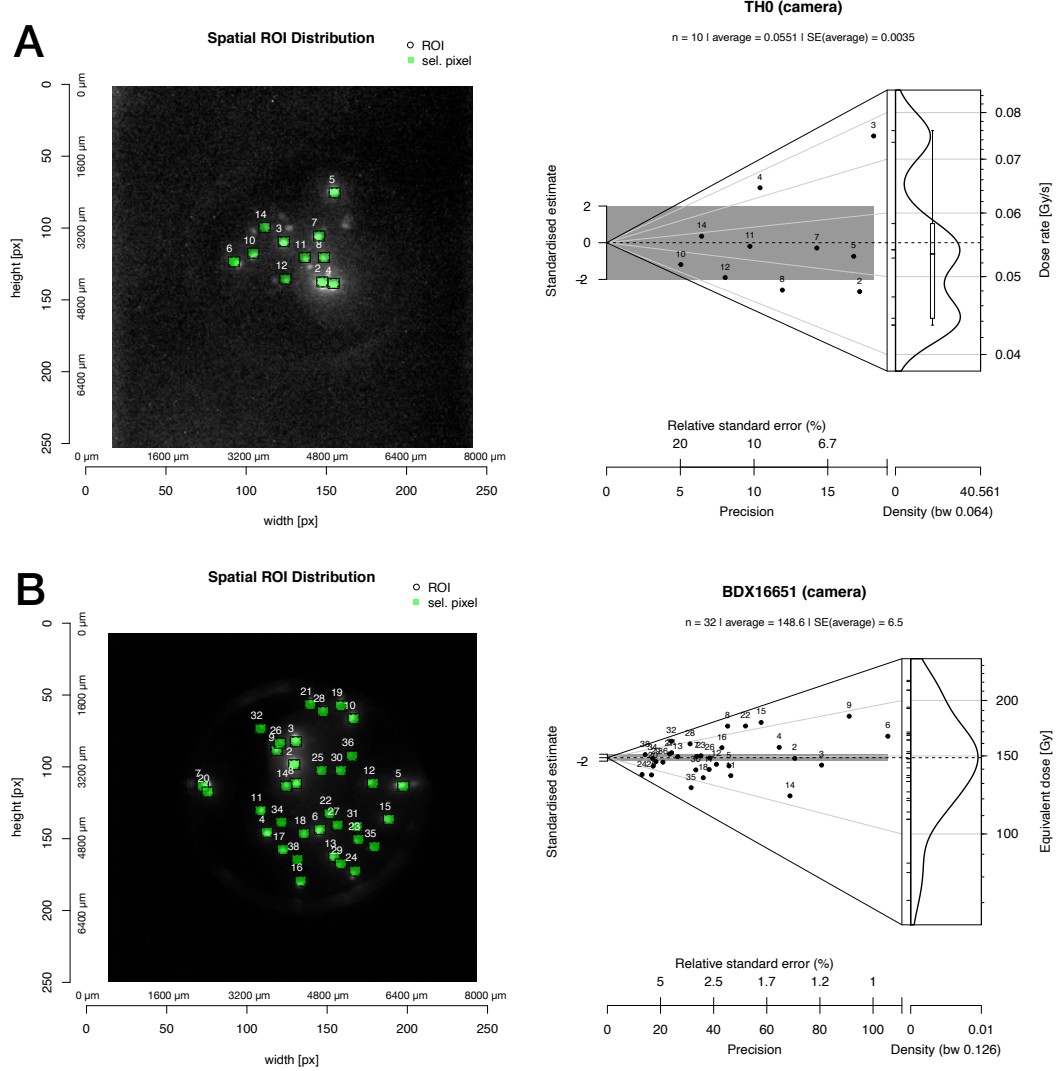

**Figure 6.** The figure shows the SR IR-RF measurement results for single grains from samples TH0 (A, top) and BDX16651 (B, bottom). For each sample, the image of one aliquot with the selected ROIs (green) is plotted on the lefthand side, and the resulting $D_e$ distribution as Abanico plot (Dietze et al., 2016) on the righthand side. Numbers (white) in the plots identify individual ROIs. The distribution for TH0 shows dose rates and the distribution for BDX16651 equivalent doses. Note: The *SR-RF* macro returns ROI images only as low-res '*.png' files. Thus for illustrative reasons, the ROI overlay images were produced manually for this manuscript from images returned by the 'SR-RF' *ImageJ* macro and R.



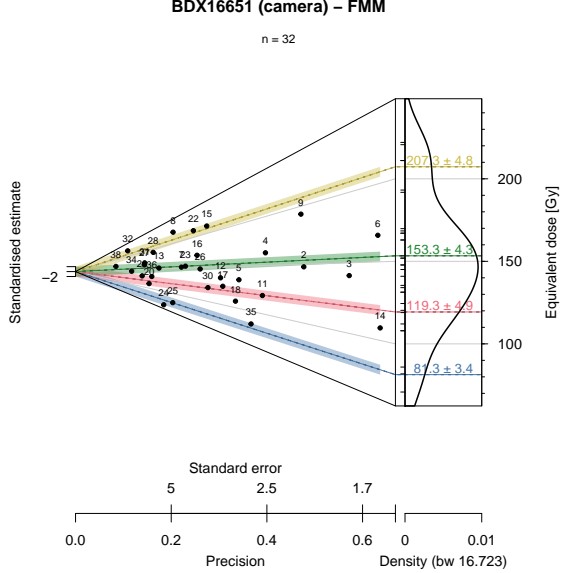

**Figure 7.** Abanico plot for sample BDX16651 with coloured polygons indicating dose components as identified by the finite mixture model. Further details see maintext.

### 3.4.2 BDX16651

We counted ca 40 grains on the analysed cup and 35 emitted light and were analysed. We discarded three grains because the R analysis indicated a bad match of $RF_{nat}$ and $RF_{reg}$. The sample shows a large $D_e$ scatter with an average $D_e$ of $148.6 \pm 6.7$ Gy (average dose and associated standard error (SE)). This value is significantly larger than the mean $D_e$ of ca 96 Gy reported by Kreutzer et al. (2018). However, in contrast to the study by Kreutzer et al. (2018), the single grain data allow further statistical treatment of the results. We applied the Finite Mixture Model (FMM, cf. Galbraith and Roberts, 2012)

using the function `Luminescence::calc_FiniteMixture()` with an assumed `sigmab` value of 0.05 (Fig. 7). The Bayesian information criterion indicated the statistically significant number of components.

We found that four-dose components can best describe the $D_e$ distribution. The lowest ca 81 Gy (blue colour, Fig. 7) contains only 10% of all grains, the 2nd component ca 26%, the 3rd ca 47% and the highest dose component ca 19% of all grains. The number varies with `sigmab` (not shown), but the dataset seems to consist of at least two dose groups (around $< 120$ Gy and

$> 120$ Gy). Assuming that the lowest dose group (Fig. 7) corresponds to the best bleached grains (leaving aside possible layer disturbance and dose rate heterogeneities) the $D_e$ of $81.3 \pm 3.4$ Gy corresponds to an IR-RF age of ca $31 \pm 5$ Gy, this is more consistent with the quartz age of $26.1 \pm 3.5$ Gy. However, the overall statistical confidence in ages based on three grains might be doubted, regardless the statistically justified number of components. Simultaneously, it appears that dose groups with higher doses than reported by Kreutzer et al. (2018) are dominant. Here more measurements would be needed to infer a statistically

robust answer.





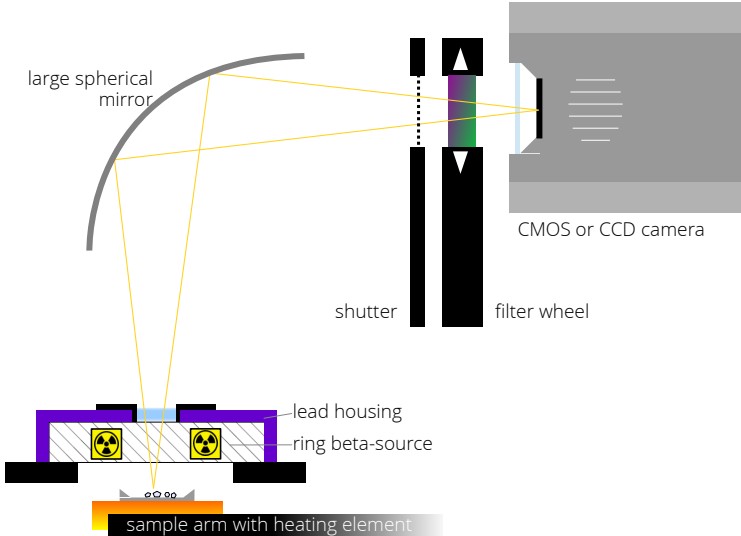

**Figure 8.** Instrumental design proposal for a dedicated SR-RF reader based on a Freiberg Instruments *lexsyg research*.

## 4 Discussion

We showed that SR-RF is technically feasible and presented first results. However, some aspects deserve critical consideration.

### 4.1 The technical dimension

It would be wishful thinking to assume that the work is finished. In comparison to PMT measurements, the number of control
parameters exploded similar to the amount of data. We tried to reduce the complexity by recommending meaningful settings and limit the number of adjustable parameters to a minimum. Still, other systems might have options we did not consider in our contribution.

Besides the technical problem we encountered with the detection chip's cooling system, we also acknowledge that our system is not perfect. A considerable improvement of image quality can be expected from a dedicated RF imaging system. We
outline a possible design for such a system in Fig. 8. A focusing off-axis parabolic mirror would allow to relocate the camera and the filters further away from the $\beta$-source and thus minimise bremsstrahlung effects and potential filter degradation. Such a mirror optic would also eliminate chromatic aberration and reduce spherical aberration given that a large mirror is chosen (spherical aberration decreases with lenses/mirrors of larger diameter). Signal cross-talk might not be eliminated but probably significantly reduced. As a camera, we propose a modern scientific CMOS camera. CMOS cameras have lower readout noise
than traditional CCD cameras although they do not support EM and hardware pixel binning.

Concerning the software, one subject for future improvements we did not implement is an advanced median filter in the image processing macro. While the current algorithm proved itself powerful in sufficiently removing speckle noise, it decreases the time resolution and deletes those pixel values which are not identified as median values. More sophisticated algorithms




deploy complex running median processes. For example, the (53H, twice)-algorithm described in Velleman (1980) would
mostly maintain time resolution while being still as potent in removing spikes. Another example: A (4253H, twice)-algorithm
Velleman (1980) would maintain the shape of the underlying RF curve while smoothing away signal spikes and much of the
Gaussian noise.

Another subject of potential improvements is the ROI assignment algorithm. The current algorithm assigns the maximum
signal pixel of the grain as the centre of the ROI, no matter if this is the middle of the grain. We suggest a subsequent algorithm
which refines the ROI centre towards an estimated grain centre. Consequently, the ROI size could be reduced without losing
signal, usually leading to higher inter-aliquot scatter. This would increase the grain separation and decrease any influence of
signal cross-talk.

Finally, while the presented software toolchain is open-source, hence freely available and open to inspections and improve-
ments, we acknowledge that the combination of three different software tools adds an additional layer of complexity. Although,
in particular, the image processing through *ImageJ* has the advantage that the data processing is transparent and available on
all platforms and independent of the particular measurement system. Furthermore, users can tap into an extensive repository of
available functions and plug-ins to record their own macros and thus adjust the image analysis with *ImageJ* without a need for
programming skills.

## 4.2  The scientific dimension

This section alludes to the scientific gain and the initially expressed hypothesis that SR IR-RF can unravel the bleaching history
of single feldspar grains.

We showed for sample TH0 that obtained source dose-rate results do not differ significantly from conventional IR-RF results
using a PMT. This observation is reassuring because it shows that the presented workflow and analysis leads to meaningful
results. However, Fig. 6A also reveals a large scatter between the individual feldspar grains ranging from $0.044\,\mathrm{Gy\,s^{-1}}$ to
$0.076\,\mathrm{Gy\,s^{-1}}$. Richter et al. (2012) reported a variation of the radiation field for our source type of only 2%. Hence, the extreme
values might result from microdosimetric effects (irradiation, cf. Mauz et al., 2020) or are related to IR-RF characteristics of
single feldspar grains. Kumar et al. (2020) reported zoning of feldspar grains linked to the geochemical composition. On some
of our images (not shown), it appears that the light is not evenly distributed over the grain surface. However, higher optical
resolutions would be required to investigate this aspect further.

Sample BDX16651 showed an even higher scatter in the equivalent doses, which is not surprising for a natural sediment
sample. While dose rate heterogeneities might add to the observed scatter (Fig. 6B) the internal K concentration of K-feldspar
(cf. Huntley and Baril, 1997), in our case contributing ca 23% to the environmental dose rate (cf. Kreutzer et al., 2018) weakens
the effect. Hence, it is more likely that the distribution reflects different bleaching histories with a lower $D_e$ component (Fig. 7)
that gives a luminescence more consistently than the quartz age. The small number of overall observations, however, does not
yet support a more robust conclusion.

Unfortunately, as already mentioned, the degraded cooling system stopped us from carrying out further experiments. More-
over, we cannot exclude the results of the two here presented cups from samples TH0, and BDX16651 remained entirely





unaffected. However, our observation was that this would be clearly visible in the IR-RF curve shapes, and it would potentially lead to vastly overestimated unrealistic results.

In the absence of such technical issues, given that our method can be tested successfully at more extensive data sets, the next logical step would be to link SR IR-RF with spectral measurements. Trautmann et al. (2000) performed spectrally resolved radiofluorescence measurements of single feldspar grains. They demonstrated that the radioluminescence emission spectra could significantly differ from grain to grain. They also showed that plagioclase grains might also emit IR-RF signals and mentioned that the separation of K-feldspar grains from other feldspar grains could not be taken for granted. Nevertheless, Trautmann
et al. (2000) concluded, that 'good' grains and 'bad' grains might be distinguishable by their spectral fingerprint. In the same year, Krbetschek et al. (2000b) showed that artificial irradiation could stimulate an RF emission centred at $700\,\mathrm{nm}$. This additional emission may interfere with IR-RF measurements. However, for the sample BDX16651, we performed brief tests with a spectrometer and did not find any indication for a potential signal interference (not shown, to be presented elsewhere).

Successive spatially resolved RF measurements at different wavelengths are possible if the measurement device deploys an
automated filter wheel. In principle, it is even possible to rotate the filter wheel during one measurement and take RF images of multiple wavelengths almost simultaneously. Nevertheless, this would require a significant software update. Still, the software framework presented in this paper may provide the basis to analyse such measurements.

Buylaert et al. (2018) unsuccessfully searched for a correlation between K concentration and the post-IR IRSL $D_e$ in single grains of K-feldspar. Recently, Kumar et al. (2020) reported a correlation of the K concentration and the IR signal measured
with cathodoluminescence. Spatially resolved RF measurements in combination with spatially resolved IRSL measurements may help to link both observations.

## 5    Conclusions

For the first time, we outlined technique and workflow for spatially resolved infrared radiofluorescence (SR IR-RF. We presented the first measurement results and a newly developed open-source software toolchain applicable manufacturer-
independent.

In contrast to routine PMT experiments, spatially resolved measurements come with more degrees of freedom that need to be taken into account, making first steps foremost a technical challenge. Our contribution detailed relevant technical parameters of the imaging system and provided application guidelines. This will allow other laboratories to repeat our work and remove significant obstacles in applying this promising method.

Tests on two K-feldspar samples showed results consistent with IR-RF measurements with a photomultiplier tube (PMT) for the sample TH0. However, our results also showed a large grain to grain scatter, requiring more attention and more future measurements. For the sample BDX16651, we identified up to four different $D_e$ components, with the lowest component resulting in an IR-RF age still older than the corresponding quartz age. This finding may indicate that this particular sample's bleaching time was insufficient to reset the natural IR-RF signal during sediment transport. However, if insufficient resetting





has affected all K-feldspar grains, it cannot be resolved by spatially resolved measurements, but it indicates the current limit of
IR-RF due to its slower signal bleachability compared to quartz.

We faced several technical issues, foremost the unstable signal background due to a camera defect. This observation demonstrates the higher complexity and potentially more error-prone technical setup than IR-RF measurements with a photomultiplier
tube. Nevertheless, we are confident that more measurements using fully functional systems can exploit the presented method's
full potential.

*Code availability.*   The for this paper developed SR-RF macro used for image processing as well as a software toolchain overview and tutorials are available at https://luminescence.de. The R package `RLumSTARR` used for an automated data processing is available at https://github.com/R-Lum/RLumSTARR. The SR-RF macro and RLumSTARR are distributed under GPL-3 licence conditions. The in this paper presented data
sets as well as the used measurement sequences and analysis scripts are available at https://doi.org/10.5281/zenodo.4395968 and make use
of the Creative Commons license (CC-BY-NC).

## Appendix A:  Estimation of the signal-to-noise ratio from the camera settings

### A1    Signal per pixel

If the signal shows no unsteadiness, inhomogeneity, non-linearity or any background signal shape, the expected signal per
image pixel $\mu_{pixel}$ (in photoelectrons e⁻) after background correction approximates to

$$\mu_{pixel} \approx t_{exposure}\, n_{bin}\, \phi_{pixel} \tag{A1}$$


where $\phi_{pixel}$ is the rate of luminescence-related photoelectrons generated in one CCD pixel in e⁻/px/s. We assume that
$\phi_{pixel}$ depends linearly on the actual photon flux emitted by the sample by an unknown constant. The other parameters are
explained and discussed in the following. Be aware that all signal and noise values in the following use the unit photoelectrons
per pixel e⁻, which is not equal to the unit counts per pixel displayed in the image data. The conversion rate between photo-
electrons and counts depends on multiple camera settings and is of just minor relevance for the signal-to-noise ratio (SNR) and
therefore not discussed here. Be also aware that if not stated otherwise, we always refer to image pixels not CCD pixels.

The **binning factor** $n_{bin}$ expresses the number of CCD pixels combined to one image pixel. Applying pixel binning improves
the signal-to-noise ratio, because the signal of the pixels is summed up, but this signal is only one time affected by readout
noise. The RF optic of the *lexsyg research* system has a lateral magnification of about M ≈ 0.6. The resulting spatial resolution
is listed in Table A1 (11/2018 @L2, IRAMAT-CRP2A).

The image **exposure time** $t_{exposure}$ (s) is user defined. It is reasonable to set the exposure time $t_{exposure}$ in dependence of
the measurements channel time $t_{channel}$ (s):





**Table A1.** Spatial resolution and binning factor in dependence of pixel binning

| Binning | $n_{bin}$ | Spatial resolution |
| :---: | :---: | :---: |
| | | ($M \approx 0.6$) |
| none | 1 | $\sim 25\,\mu$m |
| $2 \times 2$ | 4 | $\sim 50\,\mu$m |
| $4 \times 4$ | 16 | $\sim 100\,\mu$m |

    – Sequential readout (*Full Frame Mode*): $t_{exposure} = t_{channel} - t_{dead}$

– Simultaneous readout (*Frame Transfer Mode*): $t_{exposure} = t_{channel}$

If the camera runs in Full Frame Mode, any luminescence signal and trigger signal arriving during pixel shifting, pixel readout and data transmission will be lost. This is the case for measurements achieved with *LexStudio 2* (11/2018 @L2, IRAMAT-CRP2A) and results in a recommended camera dead time $t_{dead}$ (s). Table A2 shows tested and believed safe dead-time values. Within this time range, the camera will have finished image readout and transmission. Shorter dead times may

work but likely lead to lost trigger signals and therefore lost images. If the camera runs in Frame Transfer Mode (default mode for most scientific CCD imaging systems), the image shifted into a special readout section on the CCD chip after the exposure time ended. The image can be read out while the new exposure can already begin (simultaneous readout). If the exposure time is longer than the readout time (plus a little offset) no dead time is necessary.

**Table A2.** Left: CCD readout time as returned by the camera software representing theoretical lowest dead time values. Right: Experimentally derived upper dead time limits (recommended settings). For the measurement the amplifier was run in low noise mode and we used the default camera settings

| | Readout time | | | Upper dead time | | |
| :---: | :---: | :---: | :---: | :---: | :---: | :---: |
| | [s] | | | [s] | | |
| | *Binning* | | | *Binning* | | |
| **Readout rate** | *none* | $2 \times 2$ | $4 \times 4$ | *none* | $2 \times 2$ | $4 \times 4$ |
| 100 kHz | 2.13 | 0.64 | 0.21 | 2.49 | 0.85 | 0.38 |
| 1 MHz | 0.29 | 0.08 | 0.03 | 0.47 | 0.24 | 0.18 |





## A2   Noise per pixel

For each image pixel, we assume that the signal noise ($\sigma_{pixel}$) follows a normal distribution resulting from the superimposition of three independent sources:

$$\sigma_{pixel} = \sqrt{\sigma_{shot}^2 + \sigma_{dark}^2 + \sigma_{read}^2} \qquad (A2)$$

The **shot noise** ($\sigma_{shot}$) is caused by the random arrival of the photons and obeys Poisson statistics (Janesick, 2001). This noise component is independent of any camera parameter setting and only a function of the expected luminescence signal given

by the square root of Eq. A1.

The **dark shot noise** ($\sigma_{dark}$) is the shot noise of the dark current of the CCD.

$$\sigma_{dark} = \sqrt{t_{exposure}\, n_{bin}\, \phi_{dark}} \qquad (A3)$$

The **dark current signal** $\phi_{dark}$ is one of two sources of camera-internal signal backgound (besides the ADC offset, which contributes no significant noise). The dark current arises mainly from thermally released charge carriers at the CCD surface

and in the depletion region (Janesick, 2001). The dark shot noise is nearly exponentially dependent on the CCD temperature $T_{CCD}$ (°C) and its value is characteristic to each individual camera (cf. Janesick, 2001). For the sake of simplicity, we limit our discussion on the following estimation formula, derived from multiple specifications sheets of similar cameras and the *Certificate of Performance* dark charge calibration value of our specific camera (PI ProEM512B @L2, IRAMAT-CRP2A):

$$\phi_{dark} \cong 210 e^{0.144 T_{CCD}} \qquad (A4)$$

See Fig. S1 in the supplement for a plot of that equation. The dark current signal itself dissolves in the data analysis process. However, the contribution of the dark shot noise in Eq. A2 grows with the binning factor, the exposure time and especially the CCD temperature. The dark shot noise of the camera used in this contribution is estimated in Table A3.

The **readout noise** $\sigma_{read}$ is a result of the register readout. It is independent of the exposure time and the CCD temperature

and approximately independent from pixel binning. The readout noise increases with increasing readout rate. For our camera and deploying the available traditional amplifier readout modes, readout noise values are listed in Table A4.

Scientific CCD cameras of this type achieve usually a readout noise of $\sim 3\,e^-$ at 100 kHz and $\sim 5\,e^-$ at 1 MHz (as for 2018).

## A3   Signal-to-noise ratio per region of interest

The **signal-to-noise** ratio (SNR) is a common quality marker for measurement data. The SNR per image pixel is defined by

$$SNR_{pixel} = \frac{\mu_{pixel}}{\sigma_{pixel}} \qquad (A5)$$





**Table A3.** Dark shot noise per image pixel estimation in dependence of exposure time, binning factor and CCD temperature for an average Princeton Instruments ProEM512 camera.

| | Estimated dark shot noise (e⁻) | | | | | |
|---|---|---|---|---|---|---|
| | $T_{CCD} = -70\,°\mathrm{C}$ | | | $T_{CCD} = -45\,°\mathrm{C}$ | | |
| | *Binning* | | | *Binning* | | |
| **Exposure time** | *none* | $2 \times 2$ | $4 \times 4$ | *none* | $2 \times 2$ | $4 \times 4$ |
| 0.5 s | 0.1 | 0.1 | 0.3 | 0.4 | 0.8 | 1.6 |
| 1 s | 0.1 | 0.2 | 0.4 | 0.6 | 1.1 | 2.3 |
| 2 s | 0.1 | 0.3 | 0.5 | 0.8 | 1.6 | 3.2 |
| 5 s | 0.2 | 0.4 | 0.8 | 1.3 | 2.5 | 5.1 |
| 10 s | 0.3 | 0.6 | 1.2 | 1.8 | 3.6 | 7.2 |
| 20 s | 0.4 | 0.8 | 1.7 | 2.5 | 5.1 | 10.2 |
| 40 s | 0.6 | 1.2 | 2.4 | 3.6 | 7.2 | 14.4 |

**Table A4.** Readout noise dependence of readout rate using the traditional low noise amplifier. Values according to the *Certificate of Performance* of the PI ProEM512B camera at device L2, IRAMAT-CRP2A

| Readout rate | Readout noise |
|---|---|
| | [e⁻] |
| 100 kHz | 3.3 |
| 1 MHz | 7.5 |
| 5 MHz | 14.1 |

Signal per image pixel and noise per image pixel can be approximated by Eq. A1 and Eq. A2. However, of more interest than the pixel SNR is the SNR of one grain's RF signal. In terms of image processing, a grain is defined by its **region-of-interest** (ROI). It is reasonable to set the ROI diameter about 50 % larger than the average grain diameter. With the lateral magnification known (here M = 0.6) and the CCD pixel size known (here $d_{pixel} = 16\ \mu$m), measures of length can be converted into pixels and vice versa.

For our SNR approximation, we assume that the grain IR-RF signal remains on a steady level (no decay) and that all signal light reaching the CCD is gathered in the associated ROI (no light scattering). Then, the IR-RF **signal per grain** ($\mu_{grain}$) depends just on the exposure time and the emitted photon flux of the grain.

$$\mu_{grain} = t_{exposure}\phi_{grain} \tag{A6}$$



Here, $\phi_{grain}$ is the CCD signal flux per grain (in e$^-$/s), which is proportional to the emitted photon flux.

In contrast, the **signal noise per grain** ($\sigma_{grain}$) depends on the ROI size, the camera settings and the grain signal itself. The summed-up signal noise per ROI is given by:

$$\sigma_{grain} = \sqrt{\mu_{grain} + n_{ROI}\sigma_{dark}^2 + n_{ROI}\sigma_{read}^2} \tag{A7}$$

Thus, the **single-grain radiofluorescence signal-to-noise ratio** ($SNR_{grain}$) can be approximated by:

$$SNR_{grain} = \frac{t_{exposure}\phi_{grain}}{\sqrt{t_{exposure}\phi_{grain} + n_{ROI}t_{exposure}n_{bin}\phi_{dark} + n_{ROI}\sigma_{read}^2}} \tag{A8}$$

Here, $n_{ROI}$ is the number of pixels in the ROI, available in the `table.rf` file. The signal per grain $\phi_{grain}$ has to be user-defined and can be considered as the same order of magnitude as the counts per second a grain would contribute to PMT measurements. We defined an arbitrary dim grain and estimated $SNR_{grain}$ for different camera settings in Table A5. We consider a SNR of at least $SNR_{grain} > 3$ as necessary to enable sufficiently precise single-grain dating.

*Author contributions.* Dirk Mittelstraß developed the *ImageJ* image processing algorithm and investigated the image aquisition issue. Sebastian Kreutzer performed the measurements, provided the interface to R and investigated the signal cross-talk issue. Both authors helped Freiberg Instruments to solve technical issues. Both authors jointly analysed the data and contributed equally to the manuscript.

*Competing interests.* The last upgrade of the IR-RF measurement system in Bordeaux was financially supported by the Freiberg Instruments GmbH. However, the manufacturer had no part in the scientific work or this manuscript. The authors declare no further competing interests.

*Disclaimer.* The authors developed the attached and linked software tools with great care and the reader may find them useful. However, the software comes WITHOUT ANY WARRANTY; without even the implied warranty of MERCHANTABILITY or FITNESS FOR A PARTICULAR PURPOSE.

*Acknowledgements.* Camille Moreau is thanked for her work in the framework of her internship at the IRAMAT-CRP2A in 2018. Ingrid Stein and Detlev Degering are thanked for safeguarding long forgotten data treasures. Chantal Tribolo and Norbert Mercier are thanked for fruitful discussions and tremendous patience while waiting for this manuscript. The authors thank the Freiberg Instruments GmbH for their support and for suffering the noise we made. Finally, we thank Daniel Nüst for creating and maintaining this wonderful Copernicus markdown template shipped with 'rticles' (Allaire et al., 2021), which made compiling this manuscript a lot easier. This work received



**Table A5.** Decision table for best binning and channel time settings. We compare the system used in our study and a (hypothetical) similar system with comparable new camera and improved control software. Bold SNR values are related to the *high SNR* camera settings, recommend in Table 3.

| Grain settings $\phi_{grain} = 20\,\mathrm{e^- s^{-1}}$ $d_{grain} = 160\,\mu m$ | | lexsyg L2 IRAMAT-CRP2A $\sigma_{read} = 3.3\mathrm{e^-}\ (7.5\mathrm{e^-})^*$ $T_{CCD} = -45\,^{\circ}C$ Sequential readout | | | improved system $\sigma_{read} = 3\mathrm{e^-}\ (5\mathrm{e^-})^*$ $T_{CCD} = -70\,^{\circ}C$ Simultaneaous readout | | |
|---|---|---|---|---|---|---|---|
| **Desired spatial resolution:** | | **25 $\mu$m** | **50 $\mu$m** | **100 $\mu$m** | **25 $\mu$m** | **50 $\mu$m** | **100 $\mu$m** |
| **Binning:** | | *none* | $2 \times 2$ | $4 \times 4$ | *none* | $2 \times 2$ | $4 \times 4$ |
| **ROI diameter:** | | 10 | 5 | 3 | 10 | 5 | 3 |
| **Desired time resolution[1]** | **Exposure time** | Single-grain IR-RF signal-to-noise ratio | | | | | |
| **2.5 s** | 0.5 s | - | $0.1^*$ | $0.3^*$ | $0.2^*$ | $0.4^*$ | 1.0 |
| **5 s** | 1 s | $0.2^*$ | $0.4^*$ | 1.1 | $0.4^*$ | 1.4 | 2.0 |
| **10 s** | 2 s | $0.5^*$ | 1.4 | 2.3 | $0.9^*$ | 2.6 | 3.6 |
| **25 s** | 5 s | 1.6 | **4.0** | 4.6 | 3.5 | **5.8** | 7.3 |
| **50 s** | 10 s | 4.3 | 7.1 | 7.1 | 6.6 | 10.0 | 11.7 |
| **100 s** | 20 s | 8.6 | 11.4 | - | 11.9 | 16.3 | - |
| **200 s** | 40 s | 14.7 | - | - | 20.3 | - | - |

[1]Time resolution of the processed image stack, given a Image Groupe Size of $n = 5$ is parametrised.

financial support by the LaScArBx LabEx, a programme supported by the ANR - n° ANR-10-LABX-52. In 2020, while the data analysis and the manuscript were completed SK has received funding from the European Union's Horizon 2020 research and innovation programme under the Marie Skłodowska-Curie grant agreement No 844457 (CREDit). DM took this research as private endevour and did not receive any specific grant from funding agencies in the public, commercial, or not-for-profit sectors.



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
