# Peer review of "Spatially Resolved Infrared Radiofluorescence: Single-grain K-feldspar Dating using CCD Imaging"

_Geochronology, 2020_

## Referee Comment (RC2)

**Spatially Resolved Infrared Radiofluorescence: Single-grain K-feldspar Dating using CCD Imaging**

Dirk Mittelstraß1 and Sebastian Kreutzer2,3

1Independent Researcher, Berthelsdorfer Str. 13, 09599 Freiberg, Germany
 2Geography & Earth Sciences, Aberystwyth University, Wales, United Kingdom

[referee-annotated manuscript omitted]

---

## Community Comment (CC1)

**Response to reviewer #1**

Dear Anonymous Reviewer,

Before I start, first, also on behalf of Dirk Mittelstrass, I may thank the reviewer for investing the time and going through our manuscript. I very much appreciate the positive feedback and the comments (of minor nature, but doubtlessly important). To get a discussion started, I list my answers below after the reviewer's comments.

Page 6, Figure 2: how was the exposure time determined at 4.2s? Why does the channel time (5s) higher than the exposure time?

Thank you for flagging this. After reading your comment, I realised that our terminology should be more transparent and better align with the software settings the user can find in *LexStudio2*. The exposure time is the actual time the camera is recording the image, and the channel time is the time between pictures because the camera has a specific dead time and needs to process the images. The screenshot shows the actual setting chosen for the measurement and is deliberately set smaller than the channel time. The channel time itself is set in the *LexStudio2* software sequence editor, which is a little bit confusing for the user. I will discuss how to avoid this confusion with Dirk, and we will come up with a revised version of the manuscript.

Page 7: The solar simulator settings here are not the same as Frouin et al. 2015. Please check the numbers

You are right. Thank you for spotting this. The here reported solar-simulator settings are correct. Unfortunately, the reference is not well suited. It should be Frouin et al. (2017). However, while double-checking, I realised that we accidentally reported the wrong settings in the manuscript in Frouin et al. (2017). Though I believe it should not matter much, the settings reported in our manuscript were also used by Frouin et al. (2017) (I verified the original sequence files). More importantly, in our case, these settings are the same used by Kreutzer et al. (2018), and we re-measured one of those samples. Our dataset available on Zenodo (Kreutzer & Mittelstrass, 2021) contains all applied sequence files. There it is visible that we applied the settings we reported. Either way, I rephrased the manuscript text to clarify that the settings are correct.

Moreover, the UV intensity has been doubled. An increase in UV induces a higher temperature during bleaching, which can activate a shallow TL peak (at ~120C at 10C/s, Huot et al., 2015). I am therefore not convinced that:

- 1. 1hour pause is sufficient for the thermocouple to completely cool down (see Huot et al., 2015 fig. 3) and for the phosphorescence signal to completely disappear,
- 2. and that the RFreg signal truly compares with the RFnat signal (if the shallow TL peak has been indeed activated during bleaching).

This is a justified claim, and it can be tested quickly. The thermocouple in the reader, installed in the sample arm, records the bleaching temperature (here at 70 °C). If your claim is correct, we should see a significant increase in the thermocouple's temperature over the 10,000 s of bleaching. Below I, exemplary, extracted all temperatures recorded during the sample's bleaching with the manuscript's settings for the sample TH0. The R script may ease an additional inspection and show how I extracted the data. The here used dataset is part of the dataset available on Zenodo (Kreutzer & Mittelstrass, 2021).

```
library(Luminescence)
temp <- read_XSYG2R("2019-09-15_20190913_TH0_blanchi_RF70_CAL_L2.xsyg",
    verbose = FALSE,
    fastForward = TRUE</pre>
```

```
) %>%
get_RLum(curveType = "measured", drop = FALSE) %>%
get_RLum(recordType = "bleaching (NA)") %>%
set_RLum(class = "RLum.Analysis", records = .) %T>%
plot_RLum(
   combine = TRUE,
   legend.pos = "bottomright",
   ylab = "Temp. thermocouple [\u00b0C]",
   xlab = "Bleaching time [s]",
   main = "Temperature thermocouple under the SLS"
)
```

**20 Temp. thermocouple [°C] 65 60 Curve 1 Curve 2 Curve 3 55 Curve 4 Curve 5 0 6000 2000 4000 8000 10000**

**Temperature thermocouple under the SLS**

The figure shows that the thermocouple's temperature remains very stable over 10,800 s (the bleaching time), even with the higher UV power settings. The true sample (grain) temperature is likely slightly higher. However, suppose the higher power settings would have caused a temperature rise, such an increase should be visible in the plot; in particular over such a long time. Luckily, a temperature increase is not visible. Moreover, the temperature stayed at the target temperature of 70°C. Hence, I am convinced that our settings are justified and did not bias our results.

Bleaching time [s]

I wonder if the large scatter on the natural sample can be due to this variation in temperature during the IR-RF measurement procedure (due to high UV contribution) and/or insufficient pause.

See my comments above, I do not think so. The measurements results do not support any interpretation in this direction.

Page 14, line 281: what signal did you use for the Feldspar paleodose?

The feldspar palaeodose was obtained with IR-RF using the  $RF_{70}$  protocol. I will add this information to the manuscript.

Page 14, line 289: same question

We will add this information in the revised manuscript.

Page 17, figure 6. It would have been nice to see a picture of each aliquot in daylight for the readers to have an idea where the signal is coming from and see how close to each other the grains actually are. It looks like the light is coming from a much smaller area than the grain. Could you comment on that?

Unfortunately, we cannot provide such a photo. Admittedly, we discussed this during writing this manuscript, but the aliquots were already discarded because I changed the laboratory and had free cups.

Besides and in general, the answer is, yes, the light seems to originate from a smaller area. The original photos taken with the camera are ready for inspection on Zenodo (Kreutzer & Mittelstrass, 2021). There we also uploaded processed data, which include images created during the analysis. From those images, it should be easier to see where the light origin.

Page 2, line 28: there are three "and" in one sentence. Please remove one at least.

I removed two 'and's.

Page 2, line 42: (e.g. Duller and Roberts (2018)) replaced by (e.g. Duller and Roberts, 2018)

Thanks corrected.

Page 6, Figure 2, caption: change the quote marks

Thanks for spotting, this was introduced by a recent change of the software that converts our manuscript written in markdown into a PDF via  $LAT_EX$ .

Page 6, line 141: "measurements: one...."

Corrected.

Page 15, figure 5, B, the graph shows a IRreg signal measured for 8000s but line 298, it is written that the measurements were done until 10,000s.

Well spotted. The text is apparently correct, and the sample was irradiated for 10,000 s. Still, it appears that the system threw away the last image dataset of the curve. Likely the reader stopped unexpectedly. We added additional information to the figure caption and the text, we had a lot of trouble with the PC during the measurement campaign.

In caption, please add "grain diameter: 7 px"

I added the unit.

Page 15, line 295: did you use silicon to mount the grains?

Yes, I added this information to the text.

Page 18, line 357: "IR-RF age of ca 31 ....Gy" and "quartz age of 26.1... Gy", you mean "ka" I guess?

Of course, our mistake, corrected.

Sebastian Kreutzer, Aberystwyth, March 3, 2021

**References**

Frouin, M., Huot, S., Kreutzer, S., Lahaye, C., Lamothe, M., Philippe, A., Mercier, N., 2017. An improved radiofluorescence single-aliquot regenerative dose protocol for K-feldspars. Quaternary Geochronology 38, 13–24. doi:10.1016/j.quageo.2016.11.004

Kreutzer, S., Duval, M., Bartz, M., Bertran, P., Bosq, M., Eynaud, F., Verdin, F., Mercier, N., 2018. Deciphering long-term coastal dynamics using IR-RF and ESR dating: A case study from Médoc, south-West France. Quaternary Geochronology 48, 108–120. doi:10.1016/j.quageo.2018.09.005

Kreutzer, Sebastian, & Mittelstrass, Dirk. (2020). Spatially Resolved Infrared Radiofluorescence (SR IR-RF) Image Data (Version 1.0) [Data set]. Zenodo. http://doi.org/10.5281/zenodo.4395968

---

## Author Response (AR1)

**Author's responses**

**Contents**

| Response to reviewer $\#1$                                                                                                                | 2 |
|-------------------------------------------------------------------------------------------------------------------------------------------|----------|
| Response to reviewer #2                                                                                                                   | 6        |
| Responses to major comments                                                                                                               | 6        |
| Responses to remarks in the PDF                                                                                                           | 8        |
| Response to reviewer #3                                                                                                                   | 9        |
| Responses to major comments                                                                                                               | 9        |
| Responses to specific comments                                                                                                            | 11       |
| Response to the editor                                                                                                                    | 14       |
| References                                                                                                                                | 16       |
| All significant changes to the manuscript are communicated in the correspondence with the referees and t editor, listed in this document. | he       |

Dirk Mittelstrass, Freiberg and Sebastian Kreutzer, Aberystwyth, March 29, 2021

**Response to reviewer #1**

Dear Anonymous Reviewer,

Before I start, first, also on behalf of Dirk Mittelstrass, I may thank the reviewer for investing the time and going through our manuscript. I very much appreciate the positive feedback and the comments (of minor nature, but doubtlessly important). To get a discussion started, I list my answers below after the reviewer's comments.

Page 6, Figure 2: how was the exposure time determined at 4.2s? Why does the channel time (5s) higher than the exposure time?

Thank you for flagging this. After reading your comment, I realised that our terminology should be more transparent and better align with the software settings the user can find in *LexStudio2*. The exposure time is the actual time the camera is recording the image, and the channel time is the time between pictures because the camera has a specific dead time and needs to process the images. The screenshot shows the actual setting chosen for the measurement and is deliberately set smaller than the channel time. The channel time itself is set in the *LexStudio2* software sequence editor, which is a little bit confusing for the user. I will discuss how to avoid this confusion with Dirk, and we will come up with a revised version of the manuscript.

Page 7: The solar simulator settings here are not the same as Frouin et al. 2015. Please check the numbers

You are right. Thank you for spotting this. The here reported solar-simulator settings are correct. Unfortunately, the reference is not well suited. It should be Frouin et al. (2017). However, while double-checking, I realised that we accidentally reported the wrong settings in the manuscript in Frouin et al. (2017). Though I believe it should not matter much, the settings reported in our manuscript were also used by Frouin et al. (2017) (I verified the original sequence files). More importantly, in our case, these settings are the same used by Kreutzer et al. (2018), and we re-measured one of those samples. Our dataset available on Zenodo (Kreutzer & Mittelstrass, 2021) contains all applied sequence files. There it is visible that we applied the settings we reported. Either way, I rephrased the manuscript text to clarify that the settings are correct.

Moreover, the UV intensity has been doubled. An increase in UV induces a higher temperature during bleaching, which can activate a shallow TL peak (at  $\sim 120$ C at 10C/s, Huot et al., 2015). I am therefore not convinced that:

- 1. 1hour pause is sufficient for the thermocouple to completely cool down (see Huot et al., 2015 fig. 3) and for the phosphorescence signal to completely disappear,
- 2. and that the RFreg signal truly compares with the RFnat signal (if the shallow TL peak has been indeed activated during bleaching).

This is a justified claim, and it can be tested quickly. The thermocouple in the reader, installed in the sample arm, records the bleaching temperature (here at 70 °C). If your claim is correct, we should see a significant increase in the thermocouple's temperature over the 10,000 s of bleaching. Below I, exemplary, extracted all temperatures recorded during the sample's bleaching with the manuscript's settings for the sample TH0. The R script may ease an additional inspection and show how I extracted the data. The here used dataset is part of the dataset available on Zenodo (Kreutzer & Mittelstrass, 2021).

```
library(Luminescence)
temp <- read_XSYG2R("2019-09-15_20190913_TH0_blanchi_RF70_CAL_L2.xsyg",
    verbose = FALSE,
    fastForward = TRUE</pre>
```

```
) %>%
get_RLum(curveType = "measured", drop = FALSE) %>%
get_RLum(recordType = "bleaching (NA)") %>%
set_RLum(class = "RLum.Analysis", records = .) %T>%
plot_RLum(
   combine = TRUE,
   legend.pos = "bottomright",
   ylab = "Temp. thermocouple [\u00b0C]",
   xlab = "Bleaching time [s]",
   main = "Temperature thermocouple under the SLS"
)
```

**2 Temp. thermocouple [°C] 65 00 Curve 1 Curve 2 Curve 3 55 Curve 4 Curve 5 0 2000 4000 6000 8000 10000 Bleaching time [s]**

**Temperature thermocouple under the SLS**

The figure shows that the thermocouple's temperature remains very stable over 10,800 s (the bleaching time), even with the higher UV power settings. The true sample (grain) temperature is likely slightly higher. However, suppose the higher power settings would have caused a temperature rise, such an increase should be visible in the plot; in particular over such a long time. Luckily, a temperature increase is not visible. Moreover, the temperature stayed at the target temperature of 70°C. Hence, I am convinced that our settings are justified and did not bias our results.

I wonder if the large scatter on the natural sample can be due to this variation in temperature during the IR-RF measurement procedure (due to high UV contribution) and/or insufficient pause.

See my comments above, I do not think so. The measurements results do not support any interpretation in this direction.

Page 14, line 281: what signal did you use for the Feldspar paleodose?

The feldspar palaeodose was obtained with IR-RF using the  $RF_{70}$  protocol. I will add this information to the manuscript.

Page 14, line 289: same question

We will add this information in the revised manuscript.

Page 17, figure 6. It would have been nice to see a picture of each aliquot in daylight for the readers to have an idea where the signal is coming from and see how close to each other the grains actually are. It looks like the light is coming from a much smaller area than the grain. Could you comment on that?

Unfortunately, we cannot provide such a photo. Admittedly, we discussed this during writing this manuscript, but the aliquots were already discarded because I changed the laboratory and had free cups.

Besides and in general, the answer is, yes, the light seems to originate from a smaller area. The original photos taken with the camera are ready for inspection on Zenodo (Kreutzer & Mittelstrass, 2021). There we also uploaded processed data, which include images created during the analysis. From those images, it should be easier to see where the light origin.

Page 2, line 28: there are three "and" in one sentence. Please remove one at least.

I removed two 'and's.

Page 2, line 42: (e.g. Duller and Roberts (2018)) replaced by (e.g. Duller and Roberts, 2018)

Thanks corrected.

Page 6, Figure 2, caption: change the quote marks

Thanks for spotting, this was introduced by a recent change of the software that converts our manuscript written in markdown into a PDF via  $LAT_EX$ .

Page 6, line 141: "measurements: one...."

Corrected.

Page 15, figure 5, B, the graph shows a IRreg signal measured for 8000s but line 298, it is written that the measurements were done until 10,000s.

Well spotted. The text is apparently correct, and the sample was irradiated for 10,000 s. Still, it appears that the system threw away the last image dataset of the curve. Likely the reader stopped unexpectedly. We added additional information to the figure caption and the text, we had a lot of trouble with the PC during the measurement campaign.

In caption, please add "grain diameter: 7 px"

I added the unit.

Page 15, line 295: did you use silicon to mount the grains?

Yes, I added this information to the text.

Page 18, line 357: "IR-RF age of ca $31\ldots.{\rm Gy}$ " and "quartz age of 26.1... Gy", you mean "ka" I guess?

Of course, our mistake, corrected.

Sebastian Kreutzer, Aberystwyth, March 3, 2021

**Response to reviewer #2**

Dear Anonymous Reviewer,

We very much appreciate your supportive and thorough feedback. Please find our answers to your enclosed below.

**Responses to major comments**

It should be emphasised that the paper does not deal with all the issues necessary for age determination, it is focused on the reliable determination of De values by IR-RF on a set of single K-feldspar grains.

We agree. Strictly speaking, we focus only on the  $D_e$  determination. So perhaps the title of our manuscript is a bit misleading. On the other hand, for all the different luminescence-dating methods at hand (for instance, SAR OSL, pIRIR, TT-OSL, VSL etc.), it appears that the approach to determine  $D_e$ , sets the name of the dating method. It somewhat makes sense because the dose rate determination is relatively independent for all the different approaches to determine a  $D_e$ . Hence, we termed it 'dating', for which the reasons is perhaps now more understandable.

the bright spots were associated by the authors with bremsstrahlung photons from the beta source. To my knowledge, M.Krbetschek correlated this effect with energy deposition from cosmic radiation (muon component). Was this idea checked? Are there arguments against this idea?

Interesting point, worth its own study. We are sure that the speckle noise is caused almost entirely by bremsstrahlung. We know from radiation protection measurements, that the radiation level directly above the  $\beta$ -source shielding is at least 10 µSv / h (sorry, we have no particular numbers available and none of us has direct access to a lexsyg research device in the moment. Instead, we refer to Liritzis & Galloway (1990) who measured similar radiation levels for Risö and Daybreak TL readers). The CCD of the camera is located about 85 mm above the  $\beta$ -source ring. Therefore, we can assume that the radiation level at the camera is by far larger than the usual background radiation of about 0.1 µSv / h (which includes cosmic rays). Liritzis & Galloway (1990) explained in a conclusive way, supported by gamma spectroscopic measurements, that these radiation levels are caused by bremsstrahlung produced on the impact of  $\beta$ -particles into the shielding material. In addition, in camera tests without a nearby  $\beta$ -source we observed very few bright spots: Mostly zero or one spot per 5 sec exposure, seldom two spots. During RF measurements, we observe about 150 bright spots per 5 sec exposure. Thus, we conclude that the main part of the speckle noise must be caused by bremsstrahlung originating from the  $\beta$ -source module. From a data processing point of view, the cause does not matter much.

Our manuscript contained a short explanation and the reference to Liritzis & Galloway (1990) in Line 186 of the pre-print. To clarify the dominating cause of speckle noise, we moved this text fragment to section 2.1 "Equipment".

Figure 3: the before/after picture for step 2 (image alignment) requires some explanation. A blurry picture is presented as an improvement, this is not understandable at first glance.

We agree and added the following to figure caption:

Note: The grey step 2 images show the signal value differences between the median signal values of the natural dose RF images and the median signal values of the regenerated dose RF images. A homogeneous color means that the images are aligned.

Concerning the discussion about the De distribution of BDX16651: Dosimetric aspects should be more discussed. The existence of a broad De distribution does not necessarily result in a broad age distribution, if the K content varies from grain to grain. Here spectrometric methods for the K determination of single grains could help, e.g. far-red radiophosphorescence (Dütsch, C., Krbetschek, M.R., 1997. New methods for a better K-40 internal dose rate determination. Radiation Measurements 27, 377-381. and Krbetschek, M.R., Goetze. J, Irmer, G., Rieser, U. and Trautmann, T., 2002. The red luminescence emission of feldspar and its wavelength dependence on K, Na, Ca-composition. Mineralogy and Petrology 76, 167-177.)

We agree that grain-to-grain variations in the K-concentration might contribute to the scatter, and we now rephrase the paragraph and cite the study by Dütch and Krbetschek (1997). However, we should mention that Dütch and Krbetschek (1997) target the internal  $^{40}$ K dose rate contribution of feldspars, not specifically K-feldspar. IR-RF relies on the hypothesis that Pb replaces K; the defect responsible for IR-RF. Highest IR-RF intensities are expected from grains with higher K concentration (not necessarily but as a rule of thumb). Besides, we can try to estimate the impact of different K concentration using a simple thought experiment based on the dose data from sample BDX16651. Suppose the internal K-concentration varies between 8% and 14% and other dose rate component amounts to 2 Gy ka-1. Now we can expose the grains to such a setting for about 40 ka.

```
K_{concentration} <- seq(8, 14, 0.1)
```

```
theta_K <- 0.0565 ##Brennan 2003; for spherical grains self-absorption
dr_conv <- 0.7982 + 0.0185 ## Guérin et al., 2011, dose rate conversion factor
DR <- 2 + K_concentration * theta_K * dr_conv
De <- DR * 40</pre>
```

In this case the expected range of the  $D_e$  would be around 94.8 Gy to 105.8 Gy. Admittedly, this is a simplified scenario but should show that the large scatter we observed for our sample is unlikely to be caused dominantly by the internal K concentration.

Nevertheless, we rephrase the part a little bit to make sure that the reader does not get the impression we may have overlooked this aspect.

Furthermore, we realised that this section header was poorly chosen, making a distinction between a "technical" and a "scientific" dimension, while both have a scientific dimension. Hence, we renamed the section from "scientific dimension" to "application dimension".

In Sec. 4.1 the proposed mirror is called in the text "parabolic" but in the picture "spherical". What is correct? And - would it be beneficial to use an elliptical mirror with camera and sample in the focuses?

Thank you for spotting this discrepancy. We realised that regardless of the different labelling of the mirror in the figure and the main text, the best term would be the more general 'concave mirror system' because the actual setup might be more complicated. We changed the label in the figure to 'concave mirror' and rephrased line 372 - 376 to:

A system of one or two concave mirrors would allow to relocate the camera and the filters further away from the  $\beta$ -source and thus minimise bremsstrahlung effects and potential filter degradation. Such a mirror optic would also eliminate chromatic aberration and thus enable the ability to take RF images at different wavelengths without refocusing. A dedicated RF optic would also adress further optical aberrations and thus reduce signal cross-talk.

Concerning your second question: We assume you refer to an elliptically 'cutted' mirror to perform the 90° reflection (and not a mirror with elliptical surface). In this case, a lens optics to focus the light to the CCD

would still be needed. In fact, M. Krbetschek and D. Degering used such an optic in their experimental setup. Applying a lens optic would mean that the user either has to deal with chromatic aberration, and thus refocusing the camera each time the detection wavelength is changed (something we recommend for future research), or the user has to use an achromatic lens optic. Achromatic optics have usually weak transmission properties below 400 nm and would have trouble to detect the 300 nm peak of K-feldspar (see Murari et al., in press). An all-mirror optic has the advantage that one could measure RF emissions over the whole UV-to-NIR wavelength range without refocusing. A camera with UV-coating, like the camera we used, is sensitive from around 200 nm to about 1000 nm. After reconsidering our proposal, we guess that a system of two parabolic off-axis mirrors might give the best results while keeping the complexity and costs of the system within reasonable limits. But one large concave mirror like sketched in the figure might also work. However, this is an issue which should be calculated or simulated proberly during the actual development of a dedicated RF reader.

Although the main goal of the paper are technical aspects, in Sec. 4.2 the authors should give a list of problems to be solved before the method can be used for "real" dating purposes.

It is unfortunate (and it is still very frustrating for us) that after we solved all the tiny technical issues along the way, we could not show more dating results because of the degenerated cooling system. However, this should not leave the reader with the impression that SR IR-RF cannot be used for "real" dating purposes because our technical problem is limited to a particular system. We strongly believe that our method can be used for actual dating applications as it is. However, we are aware that further application tests are necessary and that future research will add improvements and refinements, like we stated in section 4.1.

When it comes down to IR-RF dating in general, we feel that our manuscript might not be the right place to discuss cause and implication of IR-RF related issues that may need additional research. Here we may refer to the manuscript by Murari et al. (in press), which we already cited in our manuscript. However, we believe that our method is a valuable tool to investigate some of these issues, like we stated in section 4.2.

Please let us know if this is what you had in mind. If not, of course, we are happy to add additional information to the manuscript if needed.

**Responses to remarks in the PDF**

Thank you very much for spotting all these minor mistakes we did not spot ourselves during the last check before the manuscript submission. We corrected all flagged issues where possible. Below our feedback to three of them because we felt that we better elaborate on it. > Table should be shifted before the statements.

To produce the PDF, we used the Copernicus  $IAT_EX$  template, which places the tables automatically. However, we will keep an eye on it to make sure that it is set correctly in the final published version; if accepted for publication.

too long for the line

This comment refers to a monospace-typed function call on page 12 that exceeds the column width. We acknowledged the problem and found it also at other pages. However, we found no easy fix for it and believe that this is a problem to be solved by the journal production office's final typesetting.

Dirk Mittelstrass, Freiberg and Sebastian Kreutzer, Aberystwyth, March 15, 2021

**Response to reviewer #3**

Dear Anonymous Reviewer,

Thank you very much for your detailed and thorough comments on our manuscript. Please find our answers listed below:

**Responses to major comments**

The paper is written clearly and well-presented but the authors are suggested to check a few grammar mistakes (such as the use of articles) and typos or misplaced text within a sentence including the type of English (American, British or others). I have pointed out some (find them below) but still, authors need to look at such errors throughout the text. My comments and general queries are as follows (P stands for page number):

We are sorry for the encountered typos. Thanks to the other two reviewers, we ironed out already a couple of those mistakes. Still, of course, this manuscript version was is not yet accessible. Besides, we tried to stick to British English.

Figure 5: do authors see an initial rise in the signal in these samples? The signals used as the examples in Figure 3 have it. Does this mean it is not there in the samples investigated here (Figure 5)? Have they checked this between the grains? How many initial channels (time) were removed in De determination?

This is a good observation! Indeed, in some curves, a slight initial rise is visible, mainly for curves from sample BDX16651. However, we do not consider this a problem. The arbitrary curve in figure 3 shows a stronger initial rise since this was observed in many natural (and partly regenerated) IR-RF curves (cf. Frouin et al., 2017, their supplement). However, again, this is an arbitrary curve, and it is not related to data measured in our manuscript. A general pattern is that the very first channel shows a lower count rate. This is likely due to the mechanical opening of the shutter. For the  $D_e$  estimation through sliding, the very first channel was discarded for all analyses.

We did not inspect specifically the variation of the initial rise between single grains since this was not in our analysis's focus, and we did not feel that this is of particular relevance in our manuscript.

Authors say they see at least four-dose components. Have they compared their dose response curves (DRCs)?

We have compared them during the routine data analysis, nothing particular was attracting our attention.

Could they share the data?

*GChron* strongly encourages authors to share their data. We followed this call. The **entire** dataset, i.e. all sequence files, raw image data and partially processed data (around 1.9 GB of data), had been submitted in parallel to a public repository Zenodo (Kreutzer and Mittelstrass, 2020) and is available under the Create Commons Licence.

The authors have not presented any data comparing DRCs within a sample. This is crucial since one would want to know the extent of variation in the DRCs (i.e. if any saturates later or not) in order to evaluate grain-dependent saturation (or dynamic range). Yes, you are right. These data are not directly presented in the manuscript. However, as mentioned above, our data are ready for inspection. The files also include processed dose-response data. We even provide the R scripts we used for the data analysis, so any reader can play with the data.

However, we did not see any particular reasons to compare the curves in the manuscript. They add nothing to the story (but again, such processed data is available for inspection to interested readers).

Do authors think that the grains which were not found bright, in reality, photons were produced but not detected due to the overall noise?

Of course, we cannot entirely exclude that some grains remained undetected because their signal was lower than the background noise. However, we have no evidence for such an effect. Grains showed sufficient IR light, or they did not show IR light at all and where too dim to be analysed.

In that case, noise can still be suppressed and therefore EMCCD is required since below a threshold detection point EMCCD is still better than sCMOS cameras. Do authors think that it is not important to look for those grains and only the bright grains (and resulting DRCs) will do?

This is an interesting point. At the beginning of our project, we took the EM mode into consideration and experimented with it. However, we rejected the use the of the EM mode for the reasons we stated in section 2.4.3. The by far worst problem is the pixel well overflow caused by bremsstrahlung spots. The image below shows how the resulting effect looks like for one single spot:

Figure 1: Arbitrary no-signal OSL image taken with 200x EM gain and 0.4 sec exposure time at a Princeton Instruments ProEM512B. The spot is either caused by a cosmic ray or by a bremsstrahlung photon from the ca. 50 cm away located beta source

You can see a signal "tail" or "streak" left-hand of the spot. In case of an IR-RF image with 5 sec exposure time, we see about 150 of bremsstrahlung spots with many of them having more or less intensive streaks. Most IR-RF signals but the brightest of grains become hard to observe in such a case. Also, we found the image processing far more challenging then because our median grouping algorithm ceased to work proberly. We are aware that we can decrease the spot density by decreasing the exposure time. However, then we would also decrease the SNR because of the less accumulated signal before readout. Please be aware that the readout noise is still significant in EM mode (32 e- at 5 MHz according to the Certificate of Performance). In conclusion, we achieved far better images with the conventional readout mode (EM mode turned off) as

we did not observed any pixel well overflows. Also the readout noise (3.2 e- at 100 kHz) is still low enough to ensure the detection of grains with a photon flux of just a few photons per second.

Regarding sCMOS cameras: A modern sCMOS camera can reach better SNR-per-pixel levels than an EMCCD camera at conventional readout. Scientific back-illuminated CMOS cameras have a readout noise of about 1 e- while achieving about the same quantuum efficiency than back-illuminated EMCCD cameras. In addition, sCMOS cameras offer higher image resolution and a larger field of view and thus enable a higher magnification. Also, sCMOS cameras are cheaper. We are quite sure that we would choose a sCMOS camera (but one with large pixels) if we would build a RF reader today. Nonetheless, regarding the maximum achievable sensitivity, an EMCCD camera is probably still better, provided the appearance of bremsstrahlung spots can be reduced to a sufficiently low level.

We clarified our statement regarding bremsstrahlung spot streaks in section 2.4.3.

**Responses to specific comments**

P2, line 39-40: I really think this sentence needs a word 'from' before single grains.

Done.

P2, line 43: It is 'charged coupled device (CCD)'.

Thank you, indeed.

P3, line 67: Seems 'a' is in the wrong place. Further, I do not think it is important to expand SR again since it is already done earlier. One time explanation is enough.

We rephrased the beginning of this sentence.

P7, line 148-150: Actually, most of the solar simulator settings differ a bit when compared to Frouin et al. (2015). Therefore, it is good to mention about all of them, not just UV.

Thank you, this was also flagged by the first reviewer, we clarified this now in the text.

P8, line 178: 'a' setting?

Done.

P8, line 183: '(2) it reduces the dynamic range and the linearity of the signal acquisition'...! I think this is partially true. There is a limit to which EM gain can still be applied maintaining full well capacity. Under this limit, the EM gain can increase the dynamic range by reducing the read noise.

We do not agree. Like we stated above, EM read out increases read noise and adds excess noise. To compensate these additional sources of noise, a minimum EM gain is necessary, depending on the particular camera settings (to our knowledge and experience an EM gain of about 200x is necessary to gain a significant SNR win). An EM gain high enough to improve the low range would certainly reduce the high range threshold.

P10, line 210: 'images are combined to one image by taking the median pixel value for each pixel location...! Does this mean the total no. of images are reduced by the group size during median filtering and the measurement time of each image is summed?

That is correct, the total number of images is reduced and the measurement time is summed up. So basically, the filtering reduces the resolution on the x-axis.

P11, line 222: Authors should provide more information on interpolation methods or provide references, therefore, must expand this paragraph.

The available interpolation methods are integral parts of the *ImageJ* source code and are briefly explained in the *ImageJ user guide* (Ferraira and Rasband 2012). They refer to Burger and Burge (2008) for further details on the **bicubic** interpolation. On the **bilinear** interpolation, no further references are given. We feel that details on the interpolation method are beyond scope of our contribution. Hence, we added just a reference to Ferraira and Rasband (2012) to the paragraph.

P14, line 290, The sigma here represents error?

We clarified that we quote arithmetic mean  $\pm$  standard deviation.

P15, Figure 5: Good to use unit i.e. px for diameter.

Added.

P16, line 329: define sigma\_m.

Well spotted, this might indeed not be intelligible to readers unfamiliar with the cited work. We now clarify that  $\sigma_m$  refers to the intrinsic overdispersion.

P17, Figure 6: Is it possible to re-number the grains 1-10? As the text says only 10 grains were selected.

Yes, this would be possible, but then it might become more confusing to readers who want to reproduce our results with our original dataset (which is wanted). The numbers are assigned by the algorithm, and relabelling would somewhat disconnect the automated ROI selection from the final results. Therefore we prefer to keep the numbering scheme as shown in the figure. Hopefully, this makes some sense.

Further, It would be good to keep the horizontal label within the image's boundaries.

The scales are accurate although it looks otherwise. The images had been cropped in order to avoid some issues in the presentation. We clarified the caption text and noted the cropping

P18, line 356-357: The unit will be in years, not Gy.

Done.

P19, line 371: It would be good to cite any work relevant to the potential degradation of the filter under ionising radiation.

Done.

P20, line 408: Not sure what effect?

We rephrased this part earlier, but we hope that it now reads intelligible.

P21, lin3 433: Close the bracket around SR IR-RF.

Done.

P23, line 481: the image 'is'.

Nothing changed here, perhaps page and line numbers were not correct?

P25, line 514: dpixel = 16  $\mu$ m – Is this value from the manufacturer?

Yes, the pixel size can be deduced from the data sheet of the camera.

P25, line 520: Luminescence signal (phi\_grain) per grain is basically the sum of counts from each pixel in an ROI (that has a fixed number of pixels), isn't it? The difference between eq A1 and A6 lies between how phi\_grain is chosen: per pixel or in an ROI. am I correct? I think the sum needs to be mentioned.

Yes, this part is misleading. We expanded and clarified it.

Dirk Mittelstrass, Freiberg and Sebastian Kreutzer, Aberystwyth, March 23, 2021

**Response to the editor**

Dear Prof Feathers,

Thank you very much for your support and your positive response. We carefully considered all your comments, please find our detailed answer below:

Line 43 - Is it Charge Coupled Devise (Wikipedia) or Coupled Charge Devise (your usage)? Are they interchangeable?

This was a mistake on our side, and it is now corrected.

Line 50 - Image noise and signal cross talk are technical issues, not methodological ones. Then in line 51 you can say luminescence imaging "methods"

Thank you; corrected.

Lines 151-4 - Why not give the settings and say they are identical to Frouin et al. 2017, and leave it at that, rather than mention Frouin et al. 2015 and whatever contrary information might be found in Frouin et al. 2017 (which I do not see how can be erroneously reported when your paper is not published yet.)

You are right, this makes indeed more sense, and we changed it accordingly. Regarding the reporting: perhaps what the reviewer meant was that our preprint is already citable because it has a DOI. Even the paper would be rejected for publication in the GChron, the manuscript will be still retrievable under *GChron Discuss*.

Section 2.4.1 - What kind of absolute temperatures are you talking about here? Does the cooling system come with the camera or is special equipment required?

In our case no special equipment is needed, the cooling system is inbuilt and part of the camera system. The minimum CCD temperature should be at about -75°C if no degradation have occured, like at our camera. We rephrased the part to account for your comment.

Section 2.43 - Why do you need an EM-CCD camera if you do not use the EM part of it? Does an EM-CCD camera have other qualities that make it useful. Maybe a little more justification of the camera choice could be added in your description of the camera earlier.

The camera, as available in the *lexsyg system* and used by us, was purchased to serve multiple purposes. Usually, the camera alone costs around 60 kEUR (+ VAT). This means when you buy such an expensive system (you have to add  $\sim 200$  kEUR for the measurement system), you select a camera that covers a wide range of possible application, like spatially resolved OSL or TL. Of course, if we think of an IR-RF only system, a better or cheaper choice could have been made at the cost of flexibility.

To clarify that the camera can also be used for other purposes, we stated the possibility to perform TL and OSL measurements in the Equipment section and referred to Richter et al. (2013) and Greilich et al. (2015) for examples.

Line 201 - Just to clarify, is an image stack (series of images) just a collection of light intensities as the irradiation proceeds, kind of like the bin channels in PMT output, except for several grains at a time?

Yes, this would be a sensible analogy.

Table 4 - Why does a dim sample require a smaller ROI diameter than a bright one? Intuitively, I would think the opposite, but maybe I am not understanding.

The ROI diameters at the *high SNR* settings refer to binned pixels. 2x2 superpixel have the twice the length of single pixel. Therefore, the dim sample ROIs are indeed larger than the bright sample ROIs. In an earlier version of our manuscript, we provided additional information regarding this issue in the appendix. We have removed that part due to the streamlining process before submission. Now indeed some crucial information is missing. Therefore, we expanded Table 4 to account for your comment.

Line 303 - I understand the horizontal sliding to determine the equivalent dose, but what does the vertical sliding do - correcting for some kind of sensitivity change? Instead of just citing Murari et al., perhaps an added sentence to explain this.

The vertically sliding ensures that curve shapes match. Usually, the **shape** of the IR-RF curves (natural and regenerative) are very reproducible. However, their "starting" point on the y-axis may change for different reasons (sensitivity, yes, and machine-related issues). We added a few more lines.

Line 343 - How does the pixel ROI diameter compare with the actual grain size?

We listed this information in Table 4. Our experiments found that the best results were obtained for ROIs diameters slightly larger than the optimal settings.

Line 362 - How would you propose gaining a better estimate of  $\sigma_b$ ? Dose recovery?

Complicated. The  $\sigma_b$  value is very tricky to estimate. Dose recovery tests certainly help, but additional modelling on the dose rate end would be needed too. There is currently another manuscript (Mercier et al. in preparation) for submission to *GChron* that deals with this question, but this is here beyond our scope.

Line 420 - This sentence seems garbled to me. Rewrite please.

Done and rephrased.

Lined 429 - Something is missing here.

The citation was not resolved properly, fixed.

I also read the reviewer comments and your response to reviewers 1 and 2. I think you adequately addressed their concerns, but wonder concerning Reviewer 1's point about higher heat from increased UV and whether a shallow TL peak is activated if you could add a sentence or two about this in the paper. I thought your explanation was good and deserves inclusion, to some degree, in the text.

We added a few additional lines next to the bleaching settings to clarify that the increased UV power setting did not likely change the sample's temperature.

Dirk Mittelstrass, Freiberg and Sebastian Kreutzer, Aberystwyth, March 24, 2021

**References**

Brennan, B.J., 2003. Beta doses to spherical grains. Radiation Measurements 37, 299–303. doi:10.1016/S1350-4487(03)00011-8

Burger, W. and Burge, M., 2008. Digital image processing: an algorithmic introduction using Java, 1st ed., Springer, New York. https://imagingbook.com/

Dütsch, C., Krbetschek, M.R., 1997. New methods for a better internal 40K dose rate determination. Radiation Measurements 27, 377–381. doi:10.1016/S1350-4487(96)00153-9

Ferreira, T. and Rasband, W., 2012. ImageJ User Guide IJ 1.46r. https://imagej.nih.gov/ij/docs/guide/user-guide.pdf

Frouin, M., Huot, S., Mercier, N., Lahaye, C., and Lamothe, M.: The issue of laboratory bleaching in the infrared-radiofluorescence dating method, 81, 212–217, https://doi.org/10.1016/j.radmeas.2014.12.012, 2015.

Frouin, M., Huot, S., Kreutzer, S., Lahaye, C., Lamothe, M., Philippe, A., Mercier, N., 2017. An improved radiofluorescence single-aliquot regenerative dose protocol for K-feldspars. Quaternary Geochronology 38, 13–24. https://doi.org/10.1016/j.quageo.2016.11.004

Greilich, S., Gribenski, N., Mittelstraß, D., Dornich, K., Huot, S., and Preusser, F.: Single-grain dosedistribution measurements by optically stimulated luminescence using an integrated EMCCD-based system, 29, 70–79, https://doi.org/10.1016/j.quageo.2015.06.009, 2015.

Guérin, G., Mercier, N., Adamiec, G., 2011. Dose-rate conversion factors: update. Ancient TL 29, 5–9.

Kreutzer, S., Duval, M., Bartz, M., Bertran, P., Bosq, M., Eynaud, F., Verdin, F., Mercier, N., 2018. Deciphering long-term coastal dynamics using IR-RF and ESR dating: A case study from Médoc, south-West France. Quaternary Geochronology 48, 108–120. doi:10.1016/j.quageo.2018.09.005

Kreutzer, S., & Mittelstrass, D., 2020. Spatially Resolved Infrared Radiofluorescence (SR IR-RF) Image Data (Version 1.0) [Data set]. Zenodo. http://doi.org/10.5281/zenodo.4395968

Liritzis, Y., Galloway, R. B., 1990. Bremsstrahlung from a shielded beta irradiator. Journal of Radioanalytical and Nuclear Chemistry, 146, 333–345. doi.org:10.1007/BF02164236

Murari, M.K., Kreutzer, S., King, G.E., Frouin, M., Tsukamoto, S., Schmidt, C., Lauer, T., Klasen, N., Richter, D., Friedrich, J., Mercier, N., Fuchs, M., 2021. Infrared radiofluorescence (IR-RF) dating: A review. Quaternary Geochronology 101155. doi:10.1016/j.quageo.2021.101155

Richter, D., Richter, A., and Dornich, K.: lexsyg — a new system for luminescence research, 40, 220–228, https://doi.org/10.2478/s13386-013-0110-0, 2013.